# Antibodies targeting the Crimean-Congo Hemorrhagic Fever Virus nucleoprotein protect via TRIM21

Shanna S. Leventhal[1], Thomas Bisom[1], Dean Clift [2], Deepashri Rao[1], Kimberly Meade-White[1], Carl Shaia [3], Justin Murray[1], Evan A. Mihalakakos[1], Troy Hinkley [4], Steven J. Reynolds[5], Sonja M. Best [6], Jesse H. Erasmus [4], Leo C. James [2], Heinz Feldmann [1] ✉ & David W. Hawman [1] ✉

Crimean-Congo Hemorrhagic Fever Virus (CCHFV) is a negative-sense RNA virus spread by Hyalomma genus ticks across Europe, Asia, and Africa. CCHF disease begins as a non-specific febrile illness which may progress into a severe hemorrhagic disease with no widely approved or highly efficacious interventions currently available. Recently, we reported a self-replicating, alphavirus-based RNA vaccine that expresses the CCHFV nucleoprotein and is protective against lethal CCHFV disease in mice. This vaccine induces high titers of non-neutralizing anti-NP antibodies and we show here that protection does not require Fc-gamma receptors or complement. Instead, vaccinated mice deficient in the intracellular Fc-receptor TRIM21 were unable to control the infection despite mounting robust CCHFV-specific immunity. We also show that passive transfer of NP-immune sera confers significant TRIM21-dependent protection against lethal CCHFV challenge. Together our data identifies TRIM21-mediated mechanisms as the Fc effector function of protective antibodies against the CCHFV NP and provides mechanistic insight into how vaccines against the CCHFV NP confer protection.

Crimean-Congo Hemorrhagic Fever Virus (CCHFV) was first reported in the Crimean region of the former Soviet Union and found to be antigenically similar to a virus causing illness in the Congo[1]. Since then, cases have been reported throughout southern and eastern Europe, the Middle East, India, and Asia, closely following the geographic range of *Hyalomma* genus ticks, the main reservoir of CCHFV[2,3]. Already, CCHFV is the most widespread tick-borne virus to cause disease in humans, and with climate change and global trade, there is risk for the tick to be transported and migrate into non-endemic geographic regions, placing new populations at risk for infection[4,5]. CCHFV can infect humans via the bite of infected ticks or through livestock practices such as butchering where the likelihood of being exposed to infected blood is high[2,3]. There have also been several cases of human-to-human transmission, primarily in hospital settings[6]. CCHF disease begins as a non-specific febrile illness characterized by fever, myalgia, and nausea which may progress into a severe hemorrhagic disease within 3–5 days post-symptom onset[2,3]. Case fatality rates have been reported anywhere from 5 to 70% and this large range is heavily

[1]Laboratory of Virology, Division of Intramural Research, National Institute of Allergy and Infectious Diseases, National Institutes of Health, Rocky Mountain Laboratories, Hamilton, MT 59840, USA. [2]Medical Research Council Laboratory of Molecular Biology, Cambridge CB20QH, UK. [3]Rocky Mountain Veterinary Branch, Division of Intramural Research, National Institute of Allergy and Infectious Diseases, National Institutes of Health, Rocky Mountain Laboratories, Hamilton, MT 59840, USA. [4]HDT Bio, Seattle, WA 98102, USA. [5]Laboratory of Immunoregulation, Division of Intramural Research, National Institute of Allergy and Infectious Diseases, National Institutes of Health, Bethesda, MD 20892, USA; Johns Hopkins School of Medicine, Baltimore, MD 21205, USA. [6]Laboratory of Neurological Infections and Immunity, Division of Intramural Research, National Institute of Allergy and Infectious Diseases, National Institutes of Health, Rocky Mountain Laboratories, Hamilton, MT 59840, USA. ✉e-mail: feldmannh@niaid.nih.gov; david.hawman@nih.gov

dependent on country surveillance and case reporting infrastructure[1–3]. Unfortunately, there is yet to be an effective and approved vaccine for widespread usage and therapeutic options remain limited to supportive care[2]. This is due in part to the lack of understanding CCHFV pathogenesis and interactions with the innate and adaptive immune systems. Due to this lack of countermeasures and its epidemic potential, the World Health Organization has listed CCHFV as a high priority pathogen since 2015, highlighting the need to develop effective vaccines and therapeutics.

CCHFV is a negative-sense RNA bunyavirus belonging to the family *Nairoviridae*[2,3]. Like other bunyaviruses, CCHFV has a tri-segmented genome comprised of a small (S), medium (M) and large (L) genomic segments[2,3,7]. The S segment encodes the nucle protein (NP) and a non-structural S-segment protein (NSs); the M segment encodes a glycoprotein precursor (GPC) which is processed and cleaved to release the mature glycoproteins (Gn and Gc) as well as three non-structural proteins, and the L segment encodes a large polyprotein which includes the RNA-dependent RNA-polymerase (RdRp) as well as an ovarian tumor-like (OTU) deubiquitinase domain and several regions of unknown function[2,3,7]. The CCHFV NP has multiple functions including encapsidating the viral genome, promoting genomic replication and inhibiting apoptosis[8]. Surprisingly, our group and others have shown that neutralizing antibodies against the viral glycoproteins are neither necessary nor sufficient for vaccine protection and multiple platforms expressing the CCHFV NP antigen have shown substantial protection against CCHFV in pre-clinical challenge models[9–14]. Others have shown that non-neutralizing antibodies against the viral glycoproteins or accessory protein GP38 can also protect[15,16] further demonstrating that neutralizing antibodies are not necessary for antibody-mediated protection against CCHFV. Recently, we reported a novel alphavirus-based replicon vaccine expressing the CCHFV NP (repNP) which is highly protective against a lethal, heterologous CCHFV challenge after a single dose in wild-type (WT) mice[17]. Protection with this vaccine correlates with high titers of non-neutralizing anti-NP antibodies and little-to-no cellular immunity in vaccinated mice and non-human primates (NHPs)[17,18]. However, for our repNP vaccine and other NP-based vaccines, it is unclear how vaccine-elicited antibodies against the intracellular and intravirion NP confer protection.

Here, we show that vaccine-induced NP-specific antibodies protect through the intracellular Fc receptor tripartite motif-containing protein 21 (TRIM21) in vivo and can inhibit CCHFV replication in vitro. We also show that complement and activating Fc-receptors are neither necessary nor sufficient for protection against lethal CCHFV challenge in NP-immune mice. Although extensively studied for its role in antiviral immunity against non-enveloped viruses, our work reveals a critical role for antibody and TRIM21 in protection against enveloped CCHFV, demonstrates that antibodies targeting the CCHFV NP can be a protective host response and provides mechanistic insight into how our and other NP-based vaccines moving towards human trials confer protection.

## Results

### Passive transfer of vaccine-elicited anti-NP antibodies are protective in naïve mice

Previously, we showed that our repRNA vaccine targeting CCHFV NP requires B-cells but not T-cells for protection[17]. We sought to verify the protective efficacy of vaccine-elicited NP-specific antibodies through passive transfer. Serum from repNP vaccinated mice used in the passive transfer had ~5.03 mg/mL total IgG with high titers of anti-CCHFV antibody dominated by IgG2c isotype, similar to our previous observations with protective repNP vaccinations[17] (Fig. 1a). When naïve mice were treated once or twice with 200 μL of immune sera and MAR1-5A3 to suppress type I IFN signaling[19], clinical disease was delayed with mean time-to-death (MTD) increased to 9 or 8 DPI, respectively, from 6

DPI in sham treated animals. Survival was significantly increased from 0% in sham-treated animals to ~40–50% (Fig. 1b, c) in immune sera treated animals. These data demonstrate that passive transfer of repNP induced anti-NP antibodies can confer significant, albeit partial protection.

### repNP vaccination is protective independently of Fcγ receptors, complement, and NK cells

The CCHFV NP is responsible for encapsidating viral genomes and is not thought to be present on the virion surface. However, NP on the surface of CCHFV-infected cells could be recognized by circulating antibody and lead to complement- or NK-cell activation, killing the infected cell. We probed CCHFV-infected mouse fibroblasts (L929) or human epithelial cells (A549) for CCHFV NP via immunofluorescence with the anti-NP monoclonal 9D5 or NP-immune sera. CCHFV NP was largely cytoplasmic and little-to-no NP was detected on the surface of unpermeabilized L929 cells (Supplementary Fig. 1a–d). In contrast, NP was detectable both intracellularly and on the surface of A549 cells (Supplementary Fig. 1e–h). Thus, the CCHFV NP may be present on the surface of infected cells, but this may be cell type specific. Since NP may be on the surface of infected cells and could be a target for antibody Fc-effector functions such as complement or NK-cell activation, we vaccinated mice deficient in activating Fc-receptors (FcγR−/−), the complement pathway (C3−/−), or WT mice depleted of NK cells with repNP and evaluated vaccine efficacy four weeks later (Fig. 2a, b). Compared to respective WT mice (C57BL6/J for C3−/− and NK depletion and C57BL6/NTac for FcγR−/−), FcγR−/− and the vaccinated WT mice depleted of NK-cells at time of challenge had similar antibody responses to vaccination (Supplementary Fig. 2a–c). However, C3−/− mice had slightly reduced antibody responses (Supplementary Fig. 2b),while none of the groups had significant T-cell responses against NP peptides (Supplementary Fig. 2d). Against lethal CCHFV challenge, repNP vaccination protected all FcγR−/− mice and mice depleted of NK cells from lethal disease, comparable to WT repNP vaccinated mice (Fig. 2c–f). C3−/− mice were also significantly protected by repNP vaccination although one mouse succumbed to disease (Fig. 2c, d). Although viral genome copies in C3−/− mice were significantly reduced compared to sham vaccinated animals, they were also significantly higher than WT repNP vaccinated mice in the liver and spleen (Fig. 2e). In addition, infectious virus was well controlled in most but not all C3−/− mice (Fig. 2f). The significant but slightly diminished protection measured in C3−/− mice suggests that repNP-elicited antibodies do not require complement for protection and instead suggests that complement is required for optimal antibody responses to the vaccine. Overall, repNP vaccination was immunogenic and protective in all vaccinated groups, indicating that non-neutralizing αNP antibodies protect independently of Fcγ receptors, the complement pathway, and NK cells.

### repNP vaccination is immunogenic but not protective in TRIM21−/− mice

We next investigated the role of the cytoplasmic Fc receptor, TRIM21[20]. As above, TRIM21−/− mice on the C57BL6/J background were vaccinated once with repNP and challenged with CCHFV 4 weeks later (Fig. 3a). At time of challenge, WT and TRIM21−/− mice had comparable immune responses to vaccination with robust, CCHFV-specific antibody titers of similar isotype (Supplementary Fig. 3a, b) and no significant T-cell responses to CCHFV NP peptides (Supplementary Fig. 3c). Strikingly, despite mounting comparable antibody responses to WT mice, all repNP vaccinated TRIM21−/− mice succumbed to the infection with the same MTD as sham-vaccinated animals (Fig. 3b, c). RepNP vaccinated TRIM21−/− mice evaluated on day 5p.i. showed uncontrolled viral replication in the blood, liver, and spleen comparable to sham vaccinated mice (Fig. 3d, e). Further, repNP vaccinated TRIM21−/− mice had significant pathology and CCHFV antigen in the liver and spleen

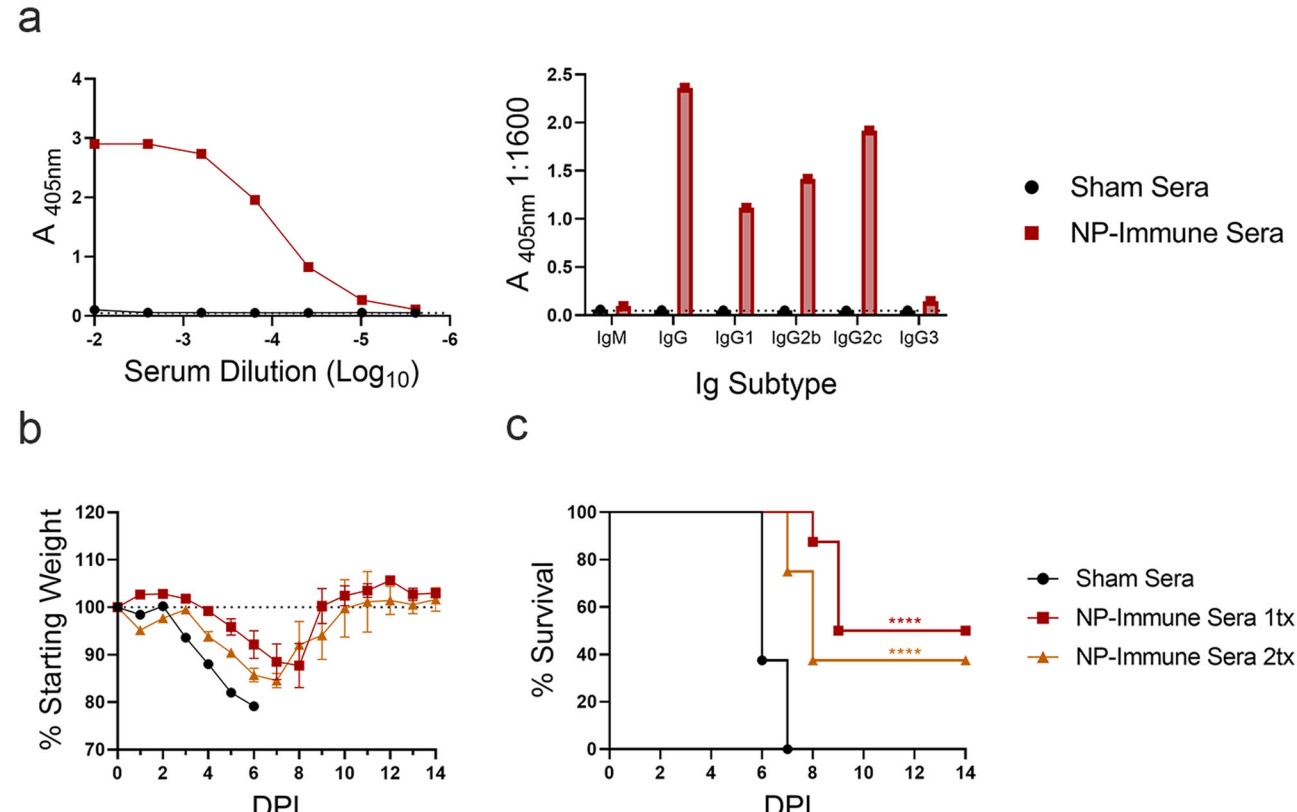

**Fig. 1 | Adoptive transfer of repNP vaccinated mouse sera increases survival in naïve CCHFV infected mice.** Sera stocks for adoptive transfer were confirmed to have CCHFV-specific antibodies via (**a**) whole virion IgG ELISA and isotype/subtype whole virion IgG ELISA. Naïve WT C57BL6/J mice were treated with sera from repNP or sham vaccinated mice on day −1 (1tx) or days 0 and +3 (2tx) relative to lethal challenge with 100 TCID$_{50}$ CCHFV strain UG3010. Mice ($N = 8$) were (**b**) weighed daily and monitored for (**c**) survival until day 14 p.i. Dashed lines indicate limit of detection. Significance was calculated using one-way ANOVA; ns $P > 0.05$, ****$P < 0.0001$. Data shown as mean plus standard deviation.

(Fig. 4a–h & Supplementary Fig. 4a–h). Cumulatively, these data demonstrate that in the absence of TRIM21, repNP-vaccinated mice are unable to control the CCHFV infection.

**Protection conferred by passively transferred sera requires TRIM21**

Together, these data suggest that anti-NP antibodies elicited by repNP vaccination require TRIM21 to confer protection. To confirm this hypothesis, we performed a passive transfer study in WT and TRIM21$^{-/-}$ mice. Additionally, we hypothesized that delivering a greater volume of immune sera and shortly prior to challenge may increase the protective efficacy of the passively transferred sera. Mice were treated with 400 µL of same immune sera characterized in Fig. 1 six-hours prior to lethal challenge with CCHFV (Fig. 5a). WT mice treated with NP immune sera had delayed weight loss and significantly improved survival with 75% (9/12) of animals surviving lethal CCHFV challenge (Fig. 5b, c). In contrast, clinical disease and survival in TRIM21$^{-/-}$ mice treated with NP-immune sera was indistinguishable from sham sera treated mice and both groups succumbed to disease with similar kinetics (Fig. 5b, c). These data indicate that protection conferred by passive transfer of NP-immune sera requires TRIM21.

**repNP vaccine-mediated protection does not require T-cells**

Our data suggest that anti-NP antibodies control CCHFV-through TRIM21-dependent mechanisms. Anti-NP antibodies against lymphochoriomeningitis virus (LCMV), a distantly related *Arenaviridae* in the Bunyaviricetes class, have been shown to control LCMV infection through TRIM21 via degradation of LCMV-NP and cross-priming of cytotoxic CD8 T-cells against the virus[21]. Although we have previously

shown that depletion of T-cells did not impact survival of vaccinated mice[17] this was evaluated in the context of mice vaccinated with repNP and an RNA expressing the CCHFV GPC. We therefore evaluated the hypothesis that, after infection, absence of TRIM21 led to diminished T-cell responses against NP and this in turn led to vaccine failure in TRIM21$^{-/-}$ mice. When we evaluated NP specific T-cell responses in the spleens of repNP-vaccinated mice by IFNγ ELISpot at day 5p.i., during peak disease and shortly before TRIM21$^{-/-}$ mice succumb, there were no significant differences between the WT and TRIM21$^{-/-}$ groups (Fig. 6a). These data suggest that NP-immune TRIM21$^{-/-}$ mice do not have defects in priming of T-cells after CCHFV infection. To further investigate the role of T-cells, we evaluated whether our repNP vaccine could protect in mice depleted of CD4 and CD8 T-cells at time of challenge (Fig. 6b) or in mice genetically deficient in CD8$^{+}$ cytotoxic T-cells (CD8$^{-/-}$). As expected, repNP-vaccinated mice depleted of both CD4 and CD8 T-cells (Fig. 6c–e) or CD8$^{-/-}$ mice (Fig. 6f, g) were completely protected against lethal CCHFV challenge. Together, our data demonstrate that anti-NP antibody mediated protection against CCHFV does not require T-cells.

**Intracellular NP antibodies can inhibit CCHFV replication**

As TRIM21 is cytoplasmic and can mediate antibody inhibition of adenovirus replication[22], we next sought to determine if cytoplasmic anti-NP antibodies could inhibit CCHFV replication by using an electroporated-antibody-dependent neutralization assay (EDNA). We adapted an established protocol in which L929 cells are electroporated in the presence of immune sera to deliver the antibody into the cytoplasm[23,24]. This assay efficiently delivered fluorescently tagged antibody to the cytoplasm of L929 cells as measured by flow cytometry

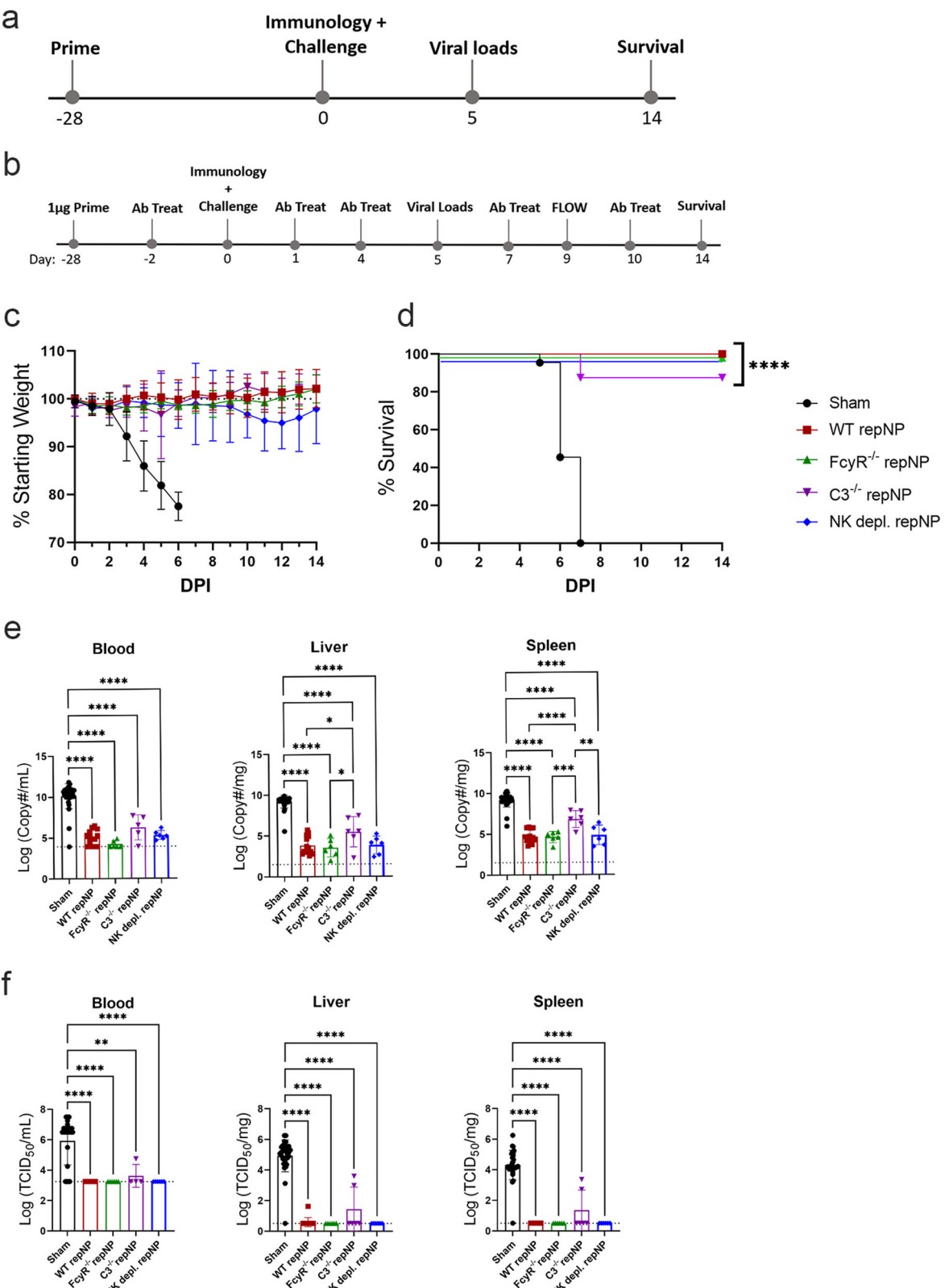

(Supplementary Fig. 5b). In the absence of electroporation, essentially no cells were positive (Supplementary Fig. 5b) indicating that electroporation is required to deliver the antibody to the cytoplasm of these cells. We next electroporated WT or TRIM21$^{-/-}$ L929 cells[23] with a dilution series of -NP-immune or sham mouse sera utilized in our passive transfer studies (Figs. 1 and 5) and then infected cells at an MOI of 0.1 with our mouse-adapted strain of CCHFV (MA-CCHFV)[25]. Viral replication at 72 HPI was quantified by TCID$_{50}$ in the supernatant. Comparing median inhibitory concentrations (IC$_{50}$s) of NP-immune sera in WT or TRIM21$^{-/-}$ cells demonstrated that TRIM21 enhanced inhibition by 17-fold (Fig. 7a) suggesting TRIM21 potentiates anti-NP antibody restriction of CCHFV. At high-concentrations of NP-immune sera, we measured inhibition of CCHFV replication independently of TRIM21 (Fig. 7a), suggesting TRIM21-independent restriction of CCHFV

**Fig. 2 | repNP vaccination is efficacious in the absence of Fcγ Receptors, Complement, and NK cells.** WT C57BL6/J or B6NTac mice, FcγR[−/−], C3[−/−] and WT mice depleted of NK cells were (**a**, **b**) vaccinated with 1ug of Sham or repNP RNA on day −28 relative to lethal CCHFV challenge. **b** WT mice were depleted of NK cells on Day −2, 1, 4, 7, and 10 relative to CCHFV challenge by IP treatment with NK1.1 antibody. On D0, groups of mice were evaluated for immunological response to vaccine or treated with MAR1-5A3 antibody and infected with a lethal dose of 100 TCID$_{50}$ CCHFV strain UG3010. Mice (N = 8) were (**c**) weighed daily and monitored for (**d**) survival until day 14 post-infection (p.i.). On D5 p.i., groups of mice (N = 6) were euthanized and evaluated for (**e**) viral genome copies via qRT-PCR and (**f**) infectious virus via TCID$_{50}$ in the blood, liver, and spleen. WT repNP mice are pooled C57BL/6 and B6NTac mice vaccinated with repNP RNA. Sham mice are pooled C57BL/6, B6NTac, FcγR[−/−], and C3[−/−] mice vaccinated with Sham RNA. Dashed lines indicate limit of detection. Significance was calculated using one-way ANOVA; ns P > 0.05, *P < 0.05, **P < 0.01, ***P < 0.001, ****P < 0.0001. Data shown as mean plus standard deviation.

by anti-NP antibody. Nonetheless, even at the highest concentration there was a resistant fraction of virus in the TRIM21[−/−] cells (Fig. 7a). In addition to extensive evaluations in mice, we have shown significant efficacy of both DNA- and RNA-based NP-expressing vaccines in non-human primates (NHPs)[13,18], suggesting this protective effect is not limited to our mouse models. We therefore used this assay to evaluate whether serum from cynomolgus macaques vaccinated with our repNP vaccine could also inhibit CCHFV. Serum from six cynomolgus macaques was collected prior to vaccination (pre-immune) or after prime-boost with our repNP vaccine (NP-immune) and pooled. We confirmed CCHFV-specific antibody responses by ELISA (Fig. 7b). Similar to our mouse sera, NP-immune sera from cynomolgus macaques electroporated into L929 cells significantly inhibited MA-CCHFV (Fig. 7c) and inhibition was enhanced by TRIM21 with a 3-fold increase in IC$_{50}$ in WT compared to TRIM21[−/−] cells (Fig. 7c). Inhibition was not observed in cells electroporated with sham or pre-immune sera (Fig. 7a, c). Furthermore, this immune sera from NHPs could induce close proximity between TRIM21 and NP in living cells (Supplementary Fig. 6a–b) suggesting that NP-specific antibodies coordinate a complex between TRIM21 and NP when present in the cytoplasm. Lastly, the CCHFV NP is often the immunodominant target of humoral immunity upon infection of naïve humans and the presence of NP antibody is frequently used for CCHFV diagnostics[26]. Therefore, we also evaluated serum from humans seropositive and negative for CCHFV to determine if antibody elicited by natural infection could inhibit in our assay (Fig. 7d). Similar to serum from vaccinated mice and cynomolgus macaques, serum from humans exposed to CCHFV and positive for NP-specific antibody inhibited CCHFV in our EDNA and inhibition was potentiated by TRIM21 with a > 6-fold increase in IC$_{50}$ when TRIM21 was present. Together, these data suggest that NP-specific antibody, elicited by vaccination or natural infection can inhibit CCHFV replication if present in the cytoplasm of infected cells and that this inhibition is potentiated by TRIM21.

### Anti-NP antibody mediated restriction of CCHFV does not require type I interferon signaling

TRIM21 can promote innate immune responses directly or upon degradation of viral capsids exposing viral genomes to nucleic acid sensors such as RIG-I-like receptors (RLRs)[27,28]. Our CCHFV challenge studies were conducted in mice treated with MAR1-5A3 to block type I IFN signaling and in which we still measured TRIM21-dependent protection, suggesting type I IFN production is not required for protection. However, as MAR1-5A3 treatment is unlikely to completely abolish type I IFN signaling, to fully rule out TRIM21-mediated induction of type I IFN via exposure of viral genomes to RLRs in protection, we utilized a lethal challenge model in which wild-type mice are infected with a high-dose of mouse-adapted CCHFV[29]. We then compared efficacy of repNP vaccination against challenge in WT, TRIM21[−/−] or mice deficient in mitochondrial antiviral signaling protein (MAVS[−/−]) an essential downstream signaling molecule for RLR signaling[30]. As in our type I IFN blockaded model, repNP conferred protection against lethal MA-CCHFV challenge in WT and MAVS[−/−] mice while protection was impaired in TRIM21[−/−] mice (Supplementary Fig. 7a–c). Although we observed severe disease in repNP-vaccinated TRIM21[−/−] mice infected with MA-CCHFV as evidenced by ~15% weight loss and similar

viral loads to sham-vaccinated mice, only 40% of mice succumbed to disease (Supplementary Fig. 7a–c) suggesting that repNP and NP-specific antibody can confer TRIM21-independent protection in this challenge model. Lastly, using a luciferase-reporter plasmid in which luciferase was under an IFNb promoter in our EDNA, we failed to measure induction of type I IFN in cells with cytoplasmic NP-specific antibody, even when infected at a high MOI (MOI = 10) at either 6 or 24 HPI (Supplementary Fig. 7d). Together these data suggest that TRIM21-mediated activation of innate immunity is not necessary for restriction of CCHFV.

## Discussion

Cumulatively, our data establish a role for anti-NP antibodies in vaccine-mediated control of lethal CCHFV challenge and demonstrate these antibodies can confer protection through TRIM21-dependent mechanisms in vivo. Our data also demonstrate that cytoplasmic anti-NP antibodies can directly inhibit CCHFV replication in vitro, even in the absence of TRIM21 suggesting these antibodies may sterically interfere with necessary interactions between NP and host or viral factors. However, this restriction is enhanced by TRIM21 and at saturating amounts of vaccine-elicited antibody, in the absence of TRIM21, we detected a resistant fraction of infectious virus similar to what has been reported for TRIM21-mediated restriction of adenoviruses[22]. The role of antibody and TRIM21 in antiviral immunity has been extensively studied in the context of non-enveloped viruses such as adenoviruses[20,31,32] and TRIM21 has also been shown to mediate restriction of the enveloped LCMV by enhancing LCMV-specific CTL responses[21]. Our work extends the antiviral role of antibody and TRIM21 in control of enveloped viruses. TRIM21 is a highly conserved protein among mammals, binds to all IgG isotypes with high affinity, can bind IgM, and can even recognize IgG from heterologous species[32,33]. TRIM21 is ubiquitously expressed including in the liver, lymph nodes and spleen[34,35], key tissues for CCHFV replication and pathology[19,25,36].

In our study passive transfer of NP-immune sera conferred significant but partial protection while repNP-vaccination conferred complete protection from clinical disease. The degradation of the NP-antibody-TRIM21 complex[37] along with the high susceptibility of type I interferon deficient mice to CCHFV (median lethal dose of less than 1 infectious unit)[38,39] may explain these distinct levels of protection. In passively transferred mice, the absence of vaccine-elicited CCHFV-specific B-cells to produce more antibody could result in decline of NP-specific antibody below protective levels resulting in uncontrolled viral replication until de novo adaptive responses arise, or the mouse succumbs. In agreement, delivering an increased volume of immune sera prior to CCHFV challenge led to greater protection and suggests even higher doses as possible through monoclonal antibody therapy may confer even greater protection. Recently, a mouse monoclonal against the CCHFV NP was shown to confer protection against lethal CCHFV challenge when administered prior to infection[40]. Similar to our vaccine studies here, protection was independent of complement or activating Fc-receptors[40] suggesting this mAb may function through TRIM21-mediated mechanisms. However, while treatment of mice with anti-NP mAb 9D5 was partially protective when given prophylactically it had no efficacy given after infection[40] suggesting this mAb is unable

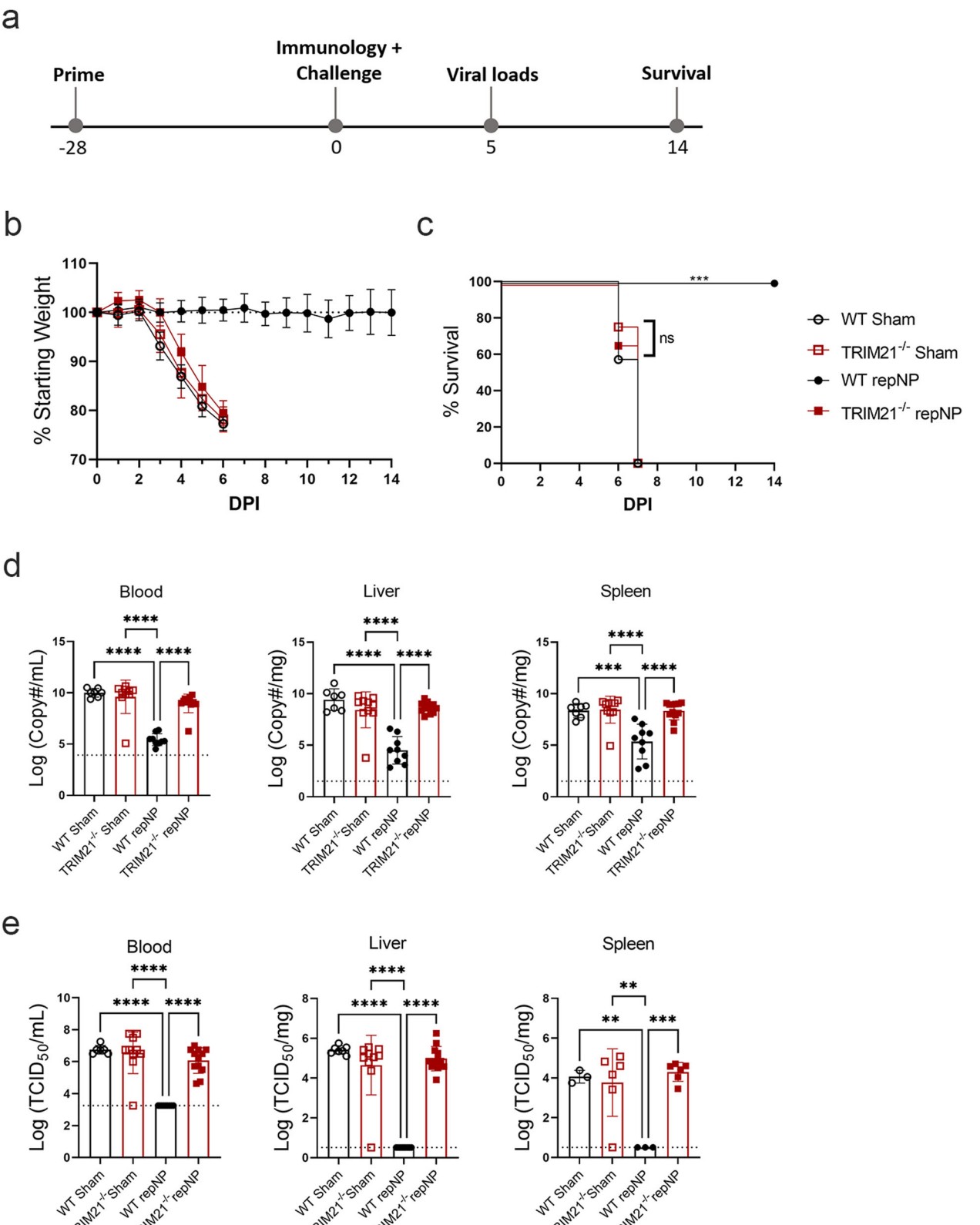

**Fig. 3 | repNP vaccination fails to protect TRIM21$^{-/-}$ mice.** WT C57BL6/J or TRIM21$^{-/-}$ mice were (**a**) vaccinated with 1ug of Sham or repNP RNA on day −28 relative to lethal CCHFV challenge. On D0, groups of mice were evaluated for immunological response to vaccine or treated with MAR1−5A3 antibody and infected with a lethal dose of 100 TCID$_{50}$ CCHFV strain UG3010. Mice (*N* = 8) were (**b**) weighed daily and monitored for (**c**) survival until day 14 p.i. On D5 p.i., groups of mice (*N* = 6) were euthanized and evaluated for (**d**) viral genome copies via qRT-PCR, and (**e**) infectious virus via TCID$_{50}$ in the blood, liver, and spleen. Dashed lines indicate limit of detection. Significance was calculated using one-way ANOVA; ns *P* > 0.05, **\*\*P* < 0.01, \*\*\**P* < 0.001, \*\*\*\**P* < 0.0001. Data shown as mean plus standard deviation. Exact *p* values: (**c**) *P* = 0.0008.

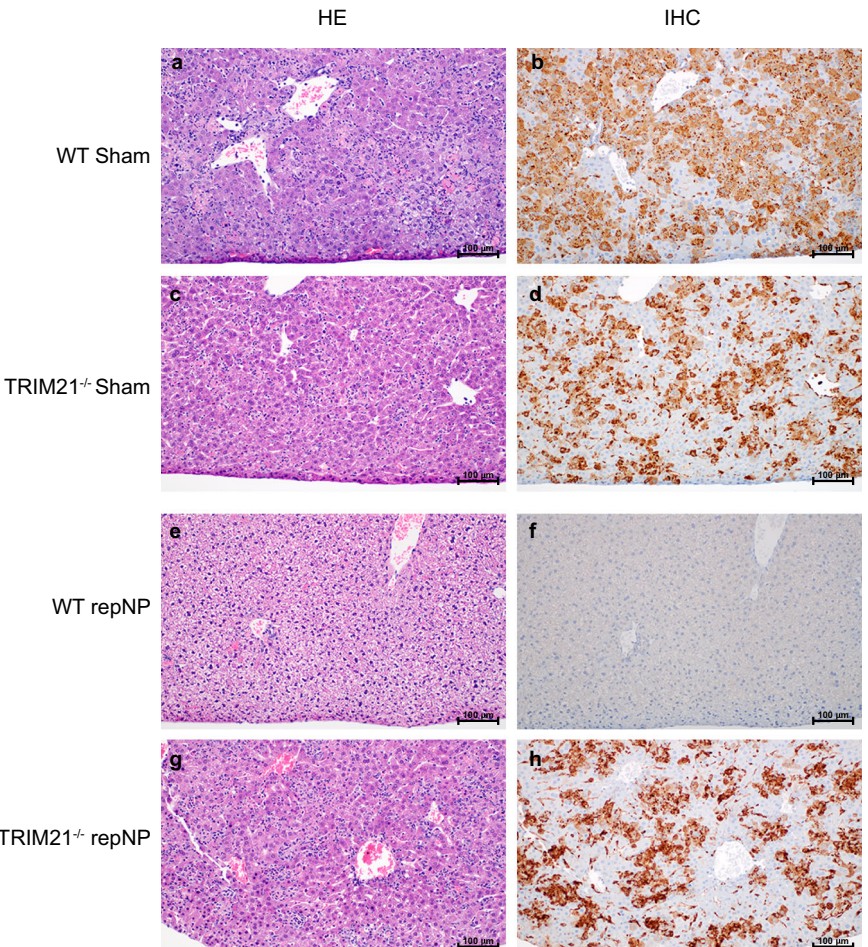

**Fig. 4 | repNP vaccination protects WT but not TRIM21−/− mice from liver pathology.** 200X magnification of liver pathology from (**a**, **b**) WT sham vaccinated mice, (**c**, **d**) TRIM21−/− sham vaccinated mice, (**e**, **f**) WT repNP vaccinated mice, and (**g**, **h**) TRIM21−/− repNP vaccinated mice with (left) HE and (right) anti-CCHF IHC reactivity staining. Images are representative of groups of $N = 6$ mice. Liver samples from sham vaccinated and TRIM21−/− repNP vaccinated mice show clusters of necrotic cellular debris multifocally dispersed throughout the hepatic plates. Sinusoidal mononuclear cells (Kupffer cells) and necrotic hepatocytes are immunoreactive. The liver samples from WT repNP vaccinated mice are normal.

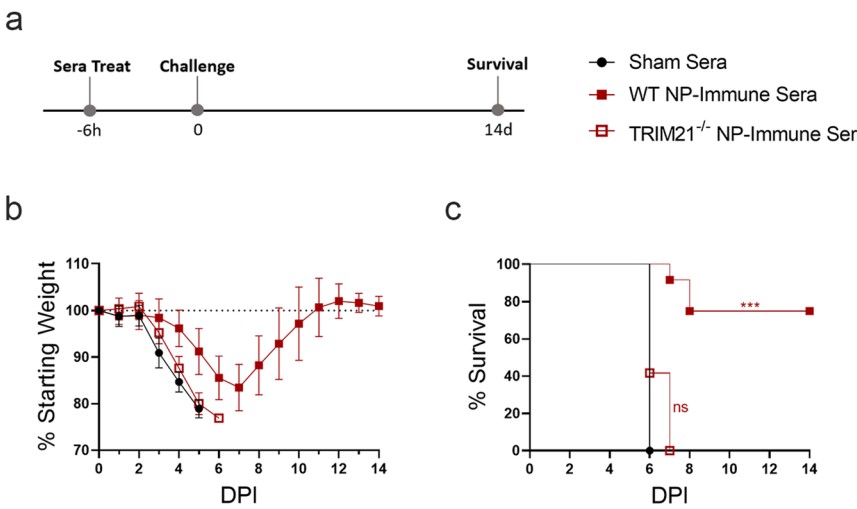

**Fig. 5 | NP-immune sera protects WT but not TRIM21−/− mice from lethal CCHFV infection.** Sera stocks for adoptive transfer were the same as described in Fig. 1. Naïve WT C57BL6/J and TRIM21−/− mice were treated with 400 ul of sham or NP-immune sera (**a**) 6 h prior to lethal challenge with 100 TCID$_{50}$ CCHFV strain UG3010. Mice ($N = 8$) were (**b**) weighed daily and monitored for (**c**) survival until day 14 p.i. Data shown as mean plus standard deviation. Significance was calculated using one-way ANOVA; ns $P > 0.05$, ***$P < 0.001$. Exact $p$-values: (**c**) $P = 0.0001$.

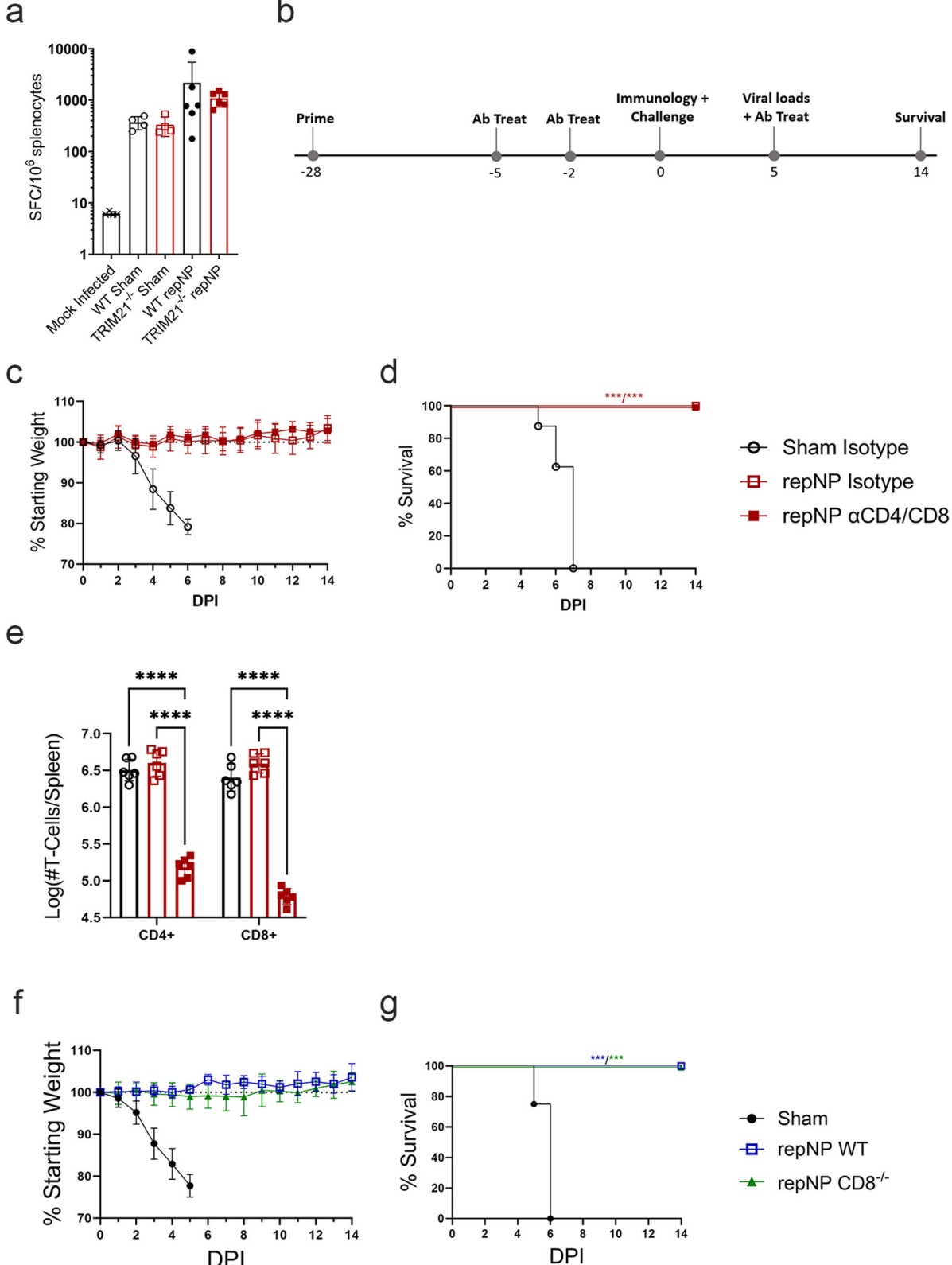

to confer protection against an established infection. Further studies are needed to determine the protective capacity and therapeutic window of NP-specific antibodies.

Although our protection studies were conducted in mice, our findings may extend to non-human primates and humans. Anti-NP antibodies were associated with protection in evaluations of a DNA-based vaccine for CCHFV in cynomolgus macaques[12,13]. In rhesus

macaques vaccinated with our repRNA vaccine, we detected antibody only against NP and levels of viral RNA in multiple tissues significantly and inversely correlated with levels of CCHFV-specific IgG[18]. We further show here that serum from repNP-vaccinated cynomolgus macaques can inhibit CCHFV replication in our EDNA, inhibition is potentiated by TRIM21 and these antibodies coordinate close molecular interaction between NP and TRIM21. The NP is an immunodominant target of host

**Fig. 6 | WT mice depleted of CD4+ and CD8 + T-cells and CD8<sup>−/−</sup> mice are protected by repNP vaccination.** On D5 p.i., groups of mice ($N = 6$) from Fig. 3 were euthanized and evaluated for cellular immune responses to infection via (**a**) IFNγ ELISpot shown as cumulative responses against peptides spanning the entire CCHFV NP (SFC: spot forming cells). For T-cell depletion study, WT C57BL6/J mice were (**b**) vaccinated with Sham or repNP RNA on day −28 relative to lethal CCHFV challenge. On days −5, −2, and +5 relative to CCHFV challenge, mice were treated with isotype or αCD4 and αCD8 antibody to deplete mice of T-cell populations. On D0, groups of mice were evaluated for immunological response to vaccine or treated with MAR1−5A3 antibody and infected with a lethal dose of 100 TCID$_{50}$

CCHFV strain UG3010. Mice ($N = 8$) were (**c**) weighed daily and monitored for (**d**) survival until day 14 p.i., On D5 p.i., groups of mice ($N = 6$) were evaluated for (**e**) depletion of CD4+ and CD8 + T-cell populations via Flow Cytometry. Antibody treatment achieved a 96.3% depletion of CD4 + T-cells and 98.5% depletion of CD8 + T-cells. In the second study, groups of WT C57BL6/J and CD8<sup>−/−</sup> mice were vaccinated and infected as above. Mice ($N = 8$) were (**f**) weighed daily and monitored for (**g**) survival until day 14 p.i. Dashed lines indicate limit of detection. Data shown as mean plus standard deviation. Significance was calculated using one-way ANOVA; ns $P > 0.05$, **$P < 0.01$, ***$P < 0.001$, ****$P < 0.0001$. Exact p-values: (**d**) both $P = 0.0002$, (**g**) both $P = 0.0002$.

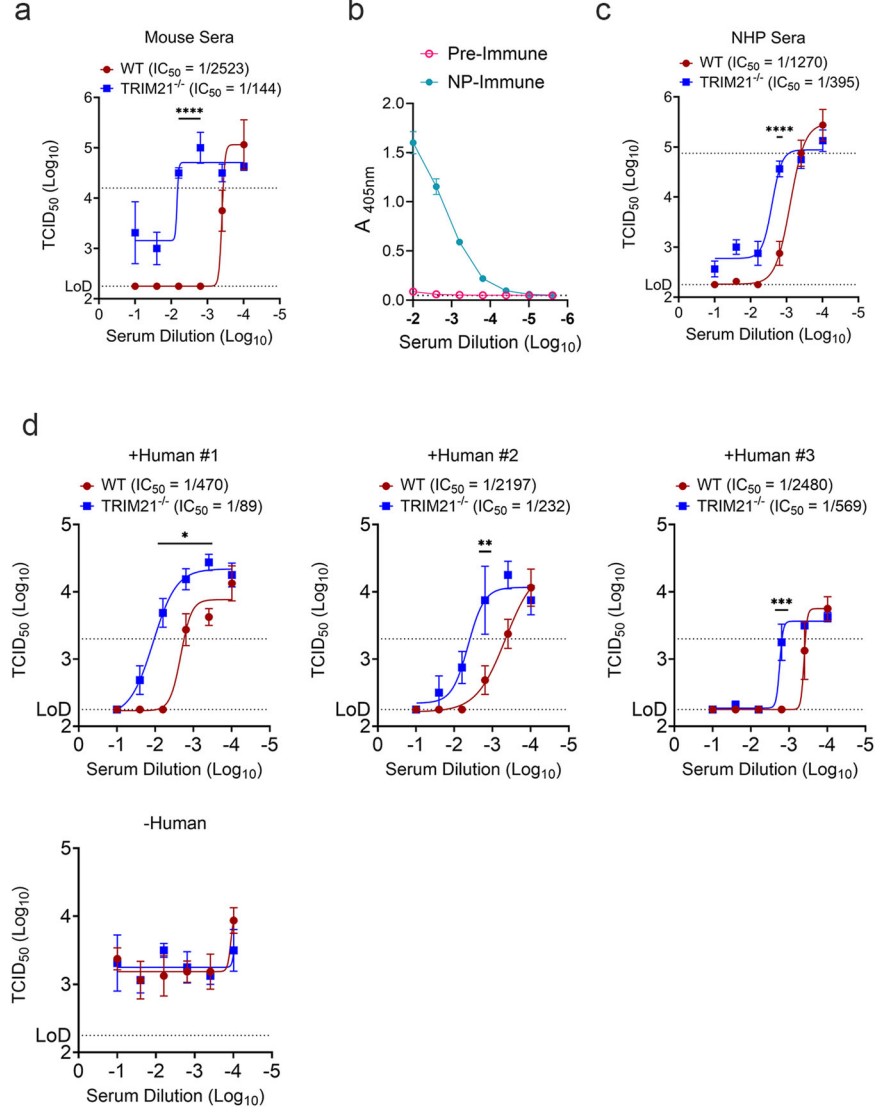

**Fig. 7 | NP-immune sera blocks CCHFV infection in vitro.** To investigate antibody-dependent intracellular neutralization (ADIN), L929 cells were electroporated with antibody and efficiency of electroporation was verified via FLOW cytometry measuring internalization of control anti-mouse AF488 antibody with and without electroporation (EP) andcells were gated by AF488- or AF488+ (Supplementary Fig. 5b). Next, L929 cells were electroporated with (**a**) mouse sham or NP-immune sera, as used in adoptive transfer studies, and infected with MA-CCHFV. Viral growth was monitored via TCID$_{50}$ 72 h p.i., Study was performed with four technical replicates. Pre-immune and NP-immune sera from cynomolgus macaques

vaccinated with our repNP vaccine was pooled and evaluated for CCHFV-specific antibody via (**b**) ELISA and then electroporated into L929 cells to assess (**c**) ADIN capacity in MA-CCHFV infected cells as above. Human sera was confirmed positive or negative for CCHFV specific antibodies via ELISA and assessed for (**d**) ADIN capacity as above. Data graphed to show inhibitory concentrations in mean plus standard deviation. Lower dashed line indicates limit of detection (LOD) of assay and upper dashed lines indicate the average TCID$_{50}$ (Log$_{10}$) of sham, pre-immune, or negative sera samples. Significance was calculated using one-way ANOVA; ns $P > 0.05$, ****$P < 0.0001$. Exact p-values: (**d**) *$P = 0.0113$; **$P = 0.0049$; ***$P = 0.0004$.

humoral immunity in multiple species including humans[41,42] and fatal cases of CCHFV often have little-to-no CCHFV-specific antibody responses prior to death[43–46]. Among survivors, neutralizing antibody responses may be low and arise well after resolution of disease[47].

Notably, appearance of anti-NP IgM antibody responses coincided with viral clearance in human patients[48]. Uncontrolled viral replication was measured in a fatal case despite detection of GPC-specific IgM and IgG but without measurable NP-specific responses[48]. We show here that

sera from CCHFV-exposed humans can inhibit CCHFV replication in our EDNA and inhibition is again enhanced by TRIM21 suggesting anti-NP antibodies that arise in humans during natural infection may contribute to control of the virus. However, as these individuals were exposed to the full CCHFV virus, we cannot exclude the possibility that antibodies against other viral proteins may have contributed to inhibition in our assay. Intriguingly, NP-based vaccines have demonstrated efficacy for other bunyaviruses such as Lassa virus[49], Rift Valley Fever Virus[50–53], Schmallenberg Virus[54] and Hantaviruses[55–57] demonstrating immunity against the bunyaviral NP is a broadly protective mechanism. However, in these studies the role of NP-specific antibody in protection was not investigated. Protective NP-specific antibodies are also not limited to bunyaviruses as passive transfer of NP-immune sera protected mice against influenza challenge[58].

An important question unanswered by our studies is how anti-NP antibodies and CCHFV NP enter the cell to interact with cytoplasmic TRIM21 and control the infection in vivo. It is likely via mechanisms distinct from those described for non-enveloped viruses[27,31]. The CCHFV NP is not reported to be a part of the viral envelope, and enveloped viruses like CCHFV undergo fusion of the viral envelope with the endosomal membrane releasing the viral genome into the cytoplasm[59,60]. This mechanism would leave any antibody-bound virion-surface exposed protein in the lumen of the endosome and shielded from cytoplasmic TRIM21. Similar to previous reports[40], in vitro we detected NP on the surface of A549 human epithelial cells but not mouse fibroblast L929 cells. Cell surface NP would expose NP to circulating antibody but it is unclear if this occurs in vivo or in all cell types infected with CCHFV. Alternatively, CCHFV NP may be released into the circulation during infection[61] which could then be bound by circulating anti-NP antibody. TRIM21 may also become exposed on the cell surface in response to type I IFNβ, cell stress or during apoptosis[32,62,63] and capture circulating or surface antibody-bound NP. Yet, how this would inhibit CCHFV replication is unclear. Additionally, although we showed that RLR-mediated sensing of viral nucleic acids and type I IFN is not necessary for protection, it is unclear what functions downstream of TRIM21 recognition of antibody-bound NP are required to restrict CCHFV. During CCHFV replication, NP binds the viral RNA genome[64] likely forming oligomers. Oligomerization of target protein helps activate TRIM21 and may lead to proteasomal degradation[65,66]. We hypothesize that this is the mechanism of TRIM21 mediated protection against CCHFV and follow-up studies will investigate the role of the proteasome in TRIM21-mediated protection.

Lastly, our data here and those recently reported[40] indicate that antibody-dependent cellular cytotoxicity and antibody-dependent complement activation are not necessary for protection by NP-specific antibody responses. In contrast, a non-neutralizing antibody against the CCHFV GP38 antigen required complement for protection[16] while a DNA-based vaccine expressing the CCHFV GPC required CD8+ T-cells, not antibody for protection[67]. However, our data do suggest complement may contribute to optimal antibody responses to our vaccine. These cumulative findings highlight that CCHFV contains multiple protective epitopes targeted by antibodies, each with distinct requirements of host immune responses for protection.

In conclusion, our study demonstrates that vaccine or infection elicited NP-specific antibodies can protect through TRIM21 in vivo. Our work expands our understanding of the antiviral function of TRIM21 against enveloped viruses and suggests that humoral immunity against the CCHFV NP, rather than just a diagnostic marker of infection, may be a key protective host response. Furthermore, the protective efficacy of passive transfer of immune sera demonstrates that anti-NP antibodies may be viable therapeutics for CCHFV, and ongoing work is evaluating this approach. Together our data provide mechanistic insight into the remarkable protection conferred by vaccine-elicited antibody directed against the CCHFV NP and inform ongoing efforts to bring crucially needed vaccines to the many people at risk of CCHFV infection.

## Methods

### Ethics
Animal work was approved by the Rocky Mountain Laboratories Institutional Animal Care and Use Committee in accordance with recommendations by the Guide for the Care and Use of laboratory Animals of the National Institutes of Health, the Office of Animal Welfare, the United States Department of Agriculture in an association for Assessment and Accreditation of Laboratory Animal Care-Accredited Facility. All work with infectious CCHFV was done at the Rocky Mountain Laboratories, NIAID, NIH in biocontainment level 4 following guidelines from the Institutional Biosafety Committee (IBC). Mice were housed in HEPA-filtered cage systems enriched with nesting material and food and water available ad libitum. Cynomolgus macaques were housed in adjoining individual primate cages that enabled social interaction, under controlled conditions of humidity, temperature and light (12-h light/12-h dark cycles). Water was available ad libitum. Animals were monitored at least twice daily and fed commercial monkey chow, treats and fruit twice a day by trained personnel. Environmental enrichment consisted of manipulanda, visual enrichment and audio enrichment. All procedures on nonhuman primates were performed by board-certified clinical veterinarians who also provided veterinary oversight of the study. Human sera samples were collected by the Rakai Community Cohort Study (parent study) which was approved by the Research and Ethics Committee of the Uganda Virus Research Institute (GC/127/19/03/709), the Uganda National Council for Science and Technology (HS-364), and the Johns Hopkins School of Medicine Institutional Review Board (IRB00204691). All participants signed written informed consent prior to enrollment. No sex/gender analysis performed.

### Vaccine
repRNA (repNP expressing the CCHFV strain Hoti nucleocapsid (NP) protein and Sham expressing an irrelevant GFP gene) was produced and complexed to LION as previously described in ref. 68.

### Animals, vaccinations, and infection
MAVS[−/−] mice on the C57BL6/J background, gift from Michael Gale, were from an in-house colony. C3[−/−] mice on the C57BL6/J background (stock #029661), and wild-type C57BL6/J (stock #00664) mice were purchased from Jackson Laboratories. Cryopreserved TRIM21[−/−] on the C57BL6/J background (stock #010724) mice were recovered at Jackson Laboratories for an in-house colony established at Rocky Mountain Laboratories, NIAID, NIH. The genotype of the initial breeders established at RML was confirmed to be TRIM21[−/−] (Transnetyx). FcγR[−/−] on the C57BL6/NTac background (Taconic stock #583) were from an in-house breeding colony and B6NTac control (stock #B6) mice were purchased from Taconic Biosciences. Except for our challenge study using MA-CCHFV (Supplementary Fig. 7), male and female mice of approximately 8-weeks age at time of vaccination or challenge in passive transfer studies were used in all studies. For our MA-CCHFV challenge study, only male mice were used as female mice do not succumb to MA-CCHFV challenge[29]. Mice were randomly assigned groups and group sizes were determined by statistical calculator to achieve a statistical power of >80% with a 95% confidence interval between groups. Vaccination consisted of a single 50 μL intramuscular injection to the left hind limb delivering 1 μg RNA as previously described in ref. 17 and appeared well tolerated. At time of challenge, mice were treated with 2.5 μg MAR1-5A3 antibody (Leinco) via a single intraperitoneal (IP) injection. Mice were then challenged IP with 100TCID$_{50}$ CCHFV strain UG3010 diluted in 100 μL media with no additives. Mice were monitored for 14 days p.i. the study endpoint. Pre-immune serum from six female cynomolgus macaques aged 8−13 years

was collected just prior to first vaccination. Animals were then vaccinated with 25 μg of repNP RNA complexed to LION via intramuscular injection. Six-weeks later animals were boosted with identical immunizations and three-weeks after boosting, serum collected for use.

## Viral stock
CCHFV strain UG3010 was originally provided by Eric Bergeron and the Centers for Disease Control and Prevention. On site, viral stock was grown up, verified, and titered as previously described[17]. Our stock of MA-CCHFV used here is identical to that described previously[29].

## Generation of Sham and NP immune Sera Stocks for passive transfer
C57BL6/J (stock #00664) mice were purchased from Jackson Laboratories. Mice were vaccinated prime-boost 4 weeks apart with repNP or sham vaccinations. 2 weeks post-second boost, mice were euthanized via terminal blood draw for collection of serum. Serum was pooled according to vaccination and confirmed to have CCHFV-specific antibodies via whole virion IgG ELISA prior to use in adoptive transfer studies (Fig. 1a).

## Depletion antibody treatments
For depletion of NK cells, wild-type C57BL6/J mice were treated with 200 μg of anti-NK1.1 (Leinco) or isotype mouse IgG2a antibody (Leinco) diluted in sterile PBS via a 100-200uL IP injection. Mice were treated with NK-depletion antibody on days -2, +1, +4, +7, and +10 relative to CCHFV challenge. The schedule was based on previous publications in the literature[69,70]. NK cell depletion was confirmed on day 9 post-infection and we observed a 15-fold reduction in NK cells (Supplementary Fig. 2e). For depletion of CD4+ and CD8 + T-cells, wild-type C57BL6/J mice were treated with 200 μg of anti-mouse CD4 clone GK1.5 (Leinco), anti-mouse CD8 clone 2.43 (Leinco), or isotype rat IgG2b (Leinco) diluted to 100uL in sterile phosphate buffered saline. Depletion efficacy was measured on day 5 p.i. and we observed > 80% x % depletion of CD4 and CD8 T-cells (Fig. 6e). Gating strategy shown in Supplementary Fig. 5a. Treatments were administered via a single IP injection.

## Immunofluorescence assay
Immunofluorescence of CCHFV-NP in L929 and A549 cells infected with MOI 1 of MA-CCHFV. At 24hpi, cells were fixed with paraformaldehyde overnight and permeabilized with saponin or unpermeabilized. CCHFV-NP was detected using repNP vaccinated mice sera (NP-Immune Sera) or the monoclonal 9D5 (BEI) applied at 1:500 and detected with a goat anti-mouse IgG conjugated to AlexaFluor488 (ThermoFisher). Cells were counterstained with Hoechst 33342. Samples were imaged using a Plan-Apochromat 63X/1.4 objective on a Zeiss laser scanning confocal microscope (LSM 880), driven by ZEN 3.0 SR (Carl Zeiss Microscopy). All images were acquired with the same settings and processed identically for publication using Adobe Photoshop v 22.5.6. Scale bar represents 10 μm. Representative fields shown.

## Enzyme-Linked Immunosorbent Assay (ELISA)
ELISA measuring antibody responses to whole virion CCHFV strain Hoti was developed in house as previously described[71]. Briefly, whole CCHFV antigen was used to coat NUNC MaxiSorp high protein-binding capacity ELISA plates (ThermoFisher) overnight before blocking with milk, incubation with mouse sera, and detection with secondary goat anti-mouse IgG antibody (Southern Biotech) and ABTS (SeraCare). For isotype/subtype ELISA, set-up was identical to whole virion ELISA however, secondary antibodies used included Goat anti-mouse IgM, IgG1, IgG2b, IgG2c, and IgG3 (Southern Biotech). Human sera was evaluated for antibody against the CCHFV NP using a dual-antigen ELISA (Innovative Diagnostics) according to manufacturer's

instructions. Positive samples were confirmed by whole-virion ELISA as described above.

## Interferon-gamma Enzyme-Linked Immunosorbent Spot (IFNγ ELISpot)
To measure CCHFV-specific T-cell responses, an IFNγ ELISpot kit (ImmunoSpot) was used with peptides spanning the entire CCHFV Hoti NP (GenScript). ELISpot was completed according to kit instructions and as previously described[17].

## Quantitative reverse-transcription PCR (qRT-PCR)
For in vivo quantification, on day 5 p.i., groups of mice were evaluated for viral genome copies in RNA from blood and tissue samples using CCHFV-specific qRT-PCR as previously described[17]. For quantification of viral RNA copies from MA-CCHFV infected cells in vitro we included a second probe with sequence CCAATGAAGTGGGGGAAGAA with a 5' FAM and 3' quencher in the reaction.

## Tissue Culture Infectious Dose 50 (TCID$_{50}$)
On day 5 p.i., groups of mice were evaluated for infectious virus in blood and tissue samples through previously described methods[17]. Briefly, blood was diluted in PBS and tissues were homogenized in 2% FBS media before serial 10-fold dilutions were performed and incubated over SW13 cells (ATCC). After 5 days, cells were evaluated for CPE. TCID50 was calculated using the Reed and Muench method.

## Electroporated Antibody Dependent Neutralization Assay (EDNA)
L929 cells (ATCC) were cultured in complete DMEM (ATCC) supplemented with 10% Fetal Bovine Serum, 50 μg/mL penicillin, and 50 μg/mL streptomycin. For electroporation, cells were prepared according to instructions provided with the Neon Transfection System MPK5000 (ThermoFisher). Briefly, cells were resuspended in Buffer R and aliquoted for each electroporation condition. 100 uL of cells was mixed with 20uL of control anti-mouse AF488 (Thermofisher) or mouse, NHP or human seraand electroporated using the Neon Transfection System 10 uL or 100 uL Kit and conditions 1400 V, 20 ms, 2 pulses. Electroporated cells were resuspended in DMEM supplemented with 10% FBS and plated in a 12-well plate. After 24 hours of recovery, cells were infected with CCHFV strains UG3010 or MA-CCHFV at an MOI of 0.1 for 1 h. Then cells were washed once with PBS and fresh, complete DMEM added. Timepoint24 or 72 HPI were taken. Viral RNA was quantified by TCID$_{50}$ as above and data presented as inhibitory concentration. Antibody delivery to cytoplasm was measured by flow cytometry 1 hour after electroporation of cells with the anti-mouse AF488-conjugated antibody (Supplementary Fig. 5b).

## Luciferase assays
WT L929 cells were electroporated using the NeonNxT Electroporation System (Thermofisher) at 1400 V, 20 ms, 2 pulses with 0.3 μg IFNβ *Firefly* luciferase and 0.2 μg pRL-TK *Renilla* control luciferase DNA constructs (Promega) with either no sera, sham sera (diluted 1:160), or repNP sera (diluted 1:160). Transfected L929 cells were seeded into every other well of tissue culture treated white-walled 96-well plates, and were incubated at 37 °C, 5% CO$_2$. The following day, transfected cells were infected with either Sendai virus (SeV) at 120 HAU/mL as a positive control for type 1 IFN activation, MA CCHFV (MOI 1 or 10), CCHFV Hoti (MOI 1), CCHFV UG3010 (MOI 1), or mock infected. Luminescence was measured at 6 and 24hpi using the Dual-Glo Luciferase Assay System (Promega) and GloMax spectrophotometer (Promega). Plotted values are RLU ratios of *Firefly*/*Renilla* luminescence normalized to RLU ratios of mock infected samples. In Supplementary Fig. 7d, values for MA-CCHFV MOI 1 and 10 infection are not significantly different and are pooled. Luciferase luminescence with Sendai Virus (SeV; Charles

River Laboratories) was used a positive control for activation of the type 1 IFN pathway.

## FLOW cytometry

To confirm NK, CD4 + T-cell and CD8 + T-cell depletions in mice treated with depletion antibodies, spleens were harvested on day 5 p.i. during planned necropsies. For a single-cell suspension, spleens were collected into RPMI-1640 media (Thermofisher) complete with 10% FBS, benzonase, and HEPES buffered saline. Spleens were crushed against and passed through a 70-micron strainer and rinsed with additional RPMI complete. Splenocytes were pelleted and resuspended in ACK lysis buffer to lyse red blood cells. Lysis was suspended by addition of cold FACS buffer (PBS + 2% FBS) and splenocytes were pelleted, washed, and used for FLOW cytometry analyses. To assess depletion of NK cells, splenocytes were stained with ZombieAqua (Biolegend), anti-mouse NK1.1 PE (Clone PK136, Biolegend), anti-mouse CD45 BUV395 (Clone 30-F11, BD Sciences), anti-mouse CD3e BV421 (Clone 145-2C11, Biolegend), anti-mouse CD11b BV786 (Clone M1/70, Biolegend), and anti-mouse CD49b APC (Clone DX5, Biolegend). To assess depletion of CD4+ and CD8 + T-cells, splenocytes were stained with ZombieAqua (Biolegend), anti-mouse CD45 BUV805 (Clone 30-F11, BD Sciences), anti-mouse CD3 PE (Clone 145-2C11, Biolegend), anti-mouse CD4 PE/Cy7 (Clone RM4.4, Biolgened), and anti-mouse CD8a AF488 (Clone 53-6.7, BD Sciences). Cells were gated to exclude debris and non-viable cells. NK cells were defined as $CD45^+Cd3^-CD11b^+CD49b^+$. T-cells were defined as $CD45^+CD3^+$ and $CD4^+$ or $CD8^+$.

## Histology

Tissues were fixed in 10% Neutral Buffered Formalin x2 changes, for a minimum of 7 days. Tissues were placed in cassettes and processed with a Sakura VIP-6 Tissue Tek, on a 12-hour automated schedule, using a graded series of ethanol, xylene, and PureAffin. Embedded tissues are sectioned at 5um and dried overnight at 42 degrees C prior to staining. Specific anti-CCHFV immunoreactivity was detected using Rabbit anti-CCHFV N IBT (Bioservices, cat#04-0011) at a 1:2000 dilution. The secondary antibody is the Immpress-VR horse anti-rabbit IgG polymer kit Vector Laboratories cat#MP-6401. The tissues were then processed for immunohistochemistry using the Discovery Ultra automated stainer (Ventana Medical Systems) with a ChromoMap DAB kit Roche Tissue Diagnostics cat#760-159.

## TRIM21:antibody:NP ternary complex assay

pGEMHE-SmBiT-TRIM21 and pGEHME-LgBiT-CCHFV(Hoti)-N-mEGFP plasmids were linearized and 5'-capped mRNA was synthesized with T7 polymerase (NEB HiScribeT7 ARCA kit; E2065S) according to manufacturer's instructions. The sequences of the plasmids are available on request. mRNA concentration was quantified using a Qubit 4 fluorometer (ThermoFisher) and RNA Broad Range assay kit (ThermoFisher; Q10211) RPE-1 cells (ATCC; CRL-4000; https://web.expasy.org/cellosaurus/CVCL_4388) were cultured in DMEM/F-12 medium (Gibco; 10565018) supplemented with 10% Calf Serum and penicillin-streptomycin at 37 °C in a 5% $CO_2$ humidified atmosphere and regularly checked to be mycoplasma-free. $8 \times 10^5$ RPE-1 TRIM21 KO cells (https://doi.org/10.1038/s41594-021-00560-2) were electroporated with a mix containing 0.25 μM each of SmBiT-TRIM21 and LgBiT-CCHFV(Hoti)-N-mEGFP mRNA and incubated for 16 h to allow protein expression. These cells were then split equally and electroporated with PBS, cyno repNP serum, cyno pre-immune serum or 1 mg/ml rabbit anti-GFP polyclonal antibody (Novus; NB600-303), transferred to a white 96-well plate (Greiner; 655083) in 100 μl volume media and incubated for 15 minutes at 37 °C. The plate was equilibrated to room temperature for 5 minutes prior to luminescence measurement using the Nano-Glo Luciferase assay system (Promega; N1130) and GloMax Discover microplate reader (Promega) according to manufacturer's instructions.

## Statistics

All statistics were done using GraphPad Prism.

## Reporting summary

Further information on research design is available in the Nature Portfolio Reporting Summary linked to this article.

## Data availability

All data underlying the figures is included in Supplementary material.

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

## Acknowledgements

We thank the Rocky Mountain Laboratories Veterinary Branch (clinical veterinary services, histology core, veterinary animal care) and the Research Technology Branch Genomics for their support of these studies. This study was supported by the Intramural Research Program of the NIAID/NIH and MCDC Grant #MCDC2204-011, D.W.H., J.H.E., H.F. Funders had no role in study design, data interpretation, or decision to publish.

## Author contributions

S.S.L., D.C., S.M.B., S.J.R., J.H.E., L.C.J., D.W.H., and H.F. conceived and designed studies. S.S.L., T.B., D.C., D.R., K.M.-W., J.M., E.A.M., T.H., L.C.J., and D.W.H. performed experiments. S.S.L., T.B., D.C., C.S., J.H.E., L.C.J., D.W.H. and H.F. performed data analysis. D.W.H., S.S.L., C.S., and H.F. wrote the manuscript. D.W.H. and S.S.L. verified the underlying data. H.F. obtained funding. All authors have read and approved final version of the manuscript.

## Competing interests

J.E. has equity interest in HDT Bio and is co-inventor on U.S. patent application no. 62/993,307 "Compositions and methods for delivery of RNA" pertaining to formulations for RNA delivery. DWH, JE and HF are inventors on U.S. patent application number 63/365,015 "Replicating RNA vaccine for Crimean-Congo hemorrhagic fever virus" regarding the repRNA for use against CCHFV. The remaining authors declare no competing interests.
