## [Peer Review file · Nature Communications]

Antibodies Targeting the Crimean-Congo Hemorrhagic Fever Virus Nucleoprotein Protect via TRIM21

Corresponding Author: Dr David Hawman

Version 0:

Reviewer comments:

Reviewer #1

(Remarks to the Author)

In this manuscript, Levental et al. demonstrated that the protective effect of an alphavirus-based vaccine expressing CCHFV NP or passive transfer of NP-immune sera functions via TRIM21 receptor. This is an interesting study, which is of significance for improving our understanding towards how these “non-neutralizing” antibodies work. However, I also have some questions with the content.

1. A major point is the absence of mechanism studies to show how the sera enter cells (independent of Fcγ receptor) and subsequently bind to intracellular TRIM21 receptor. In addition, could the authors provide evidence of either NP antibodies or antibody-virus complex could bind with TRIM21?

2. They also didn't test the downstream events--how TRIM21 confer protection against CCHFV infection. As the authors discussed in the Discussion part, there are many possible mechanisms, but none of these has been tested. I strongly recommend the authors to detect the activation of innate immune response, inflammation related factors, apoptosis, or degradation of virions/viral proteins, which may provide possible explanations for how NP antibodies work. These can be done either in vivo or in TRIM21 knock out cell lines.

3. In Fig. 6, the authors showed intracellular NP antibodies could inhibit CCHFV infection by delivering antibodies into cells via electroporation. I understand the authors want to demonstrate once the antibodies are present in the cytoplasm, they could inhibit virus infection. However, I have some concerns: 1) This experiment was conducted independent of TRIM21 (TRIM21 knock out cell lines should be used), so I don't think it can well support their above findings. 2) they showed that in the absence of electroporation, NP antibodies could not enter cytoplasm, even in the presence of CCHFV. Therefore, again, I can't imagine that in the physiological conditions, how NP antibodies permeabilize into cells? I don't think this experiment could justify the in vivo protective effect of NP antibodies.

4. I don't agree with the statement “our data...suggested the term “non-neutralizing antibody” is inaccurate to describe antibodies against CCHFV NP” (Discussion part, the fourth line). NP antibodies are protective but not neutralizing because they can't neutralize virus infection in the classical neutralizing assay in vitro. Only when they were artificially electroporated into cells, virus infection was inhibited via unknown mechanism. Please also refer a recent publication “Nucleocapsid protein-specific monoclonal antibodies protect mice against Crimean-Congo hemorrhagic fever virus (Garrison et al., Nature Communications, 2024)”, which should be cited.

Reviewer #2

(Remarks to the Author)

Reviewer #3

(Remarks to the Author)

This manuscript presents a comprehensive study of the immunological mechanisms responsible for the efficacy of an alphavirus-based RNA vaccine expressing the CCHFV nucleoprotein. CCHFV is a high priority pathogen as listed by the WHO, and development of effective vaccination is therefore critical. In this study the authors examine the importance of a number of immune cell types, receptors and proteins, and show an important role for the intracellular receptor TRIM21.

Overall this is a very thorough and robust study, with extensive use of a challenging mouse model to explore numerous hypotheses. Virus quantification by both qPCR and TCID50 across multiple organs is appreciated as being particularly rigorous. The results show a clear and convincing role for TRIM21 in vaccine-mediated protection from CCHFV. However, the precise mechanism by which TRIM21 and antibody-NP co-localize is still unclear and the authors consider this well in the discussion and explain future questions that still need to be addressed.

The difference between protection mediated by vaccination versus passively transferred antibodies is intriguing. The theory that rapid TRIM21-mediated degradation of passively transferred NP antibody accounts for failure for complete protection is an interesting one. This could be explored further by quantifying how much NP antibody persists post infection. Blood sampling daily after infection and/or analysis of blood collected at the day 5 terminal cull could be achieved to address this.

Alternatively, given the recently published study showing a role for CD8 T cells in TRIM21-dependent protection in LCMV infection, it would be interesting to see what the effect of CD8 T cell depletion has on passively transferred antibodies alone. Though the authors explore the role for T cells in vaccinated mice, this is not explored for mice that receive passively transferred antibodies. The timing of when passively transferred mice start to recover from weight loss is remarkably similar to that observed in LCMV, which in turn coincides with induction of a CD8 T cell response in naïve animals. Though it is agreed that CD8 T cells were shown not to be important for vaccine-induced recall responses, this could be important for acute naïve CD8 T cell activation and protection mediated by future monoclonal antibody therapy.

Minor comments:

Please explain what MAR1-5A3 is used for when it is first mentioned in the manuscript.

Figure 1: Error bars for first section of panel a? Label of x-axis of second panel needs updating to reflect IgM is in the panel.

Figure 2: Please clarify what is meant by 'Immunology and challenge' on day 0 (also in Fig 5).

Figure 5: statistical significance for panel a? For all IFN γ ELISPOTs please define abbreviation SFC on y axis of these graphs. What does aCD4 or CD8 do to CCHFV viral loads in the absence of a vaccine? Was the sham aCD4/CD8 experiment not performed as mice die more rapidly?

Figure 6: Please provide clearer labelling for flow cytometry panel d, which sample is electroporated? Can NP-immune sera neutralize extracellularly? It is assumed not, but comparison of intracellular and extracellular neutralization capacity would be useful. This reviewer agrees that classical neutralizing assays are clearly limited for intracellular antibody activity, but the electroporation step in the intracellular neutralization assay makes it challenging to correlate this with true value in vivo.

Discussion: TRIM21 has already been shown to be important for protection against the enveloped virus LCMV, so please update 'Our work extends the antiviral role of TRIM21 to enveloped viruses such as CCHFV'. A similar comment needs amending at the end of the discussion, but it is agreed that it is exciting to see another distinct enveloped virus protected by TRIM21.

Reviewer #4

(Remarks to the Author)

Leventhal et al. study describes a TRIM21-mediated protective mechanism of Crimean-Congo Hemorrhagic Fever Virus (CCHFV) nucleoprotein (NP) antibodies induced by a self-replicating, alphavirus-based RNA vaccine that expresses NP. The authors first demonstrate partial protection of NP immune sera in passive transfer studies. Since the NP-specific antibodies were mostly of the IgG2c isotype, the authors assessed the Fc-mediated non-neutralizing function of the induced antibodies. All the WT, Fc γ R $^{-/-}$, C3 $^{-/-}$, and NK cell-depleted mice vaccinated with the NP vaccine survived the lethal CCHFV challenge, with no significant difference in the viral loads. While comparable antibody responses in the TRIM21 $^{-/-}$ and WT vaccinated mice were observed, there was a complete loss of protection in the TRIM21 $^{-/-}$ mice. They claim NP vaccine-mediated protection is independent of T cells by performing a CD4/CD8 depletion study. The NP immune sera were not protective in a passive transfer study in TRIM21 $^{-/-}$ mice but were 75% protective in WT mice. Lastly, an in-vitro experiment suggested that the NP immune sera can significantly inhibit CCHFV replication when present inside the cytoplasm of the cell.

The study was overall well-designed. While the authors successfully demonstrate TRIM21-dependent protection of the NP antibodies induced by the vaccine, the Fc-mediated mechanism is not thoroughly demonstrated. In addition, as the paper focuses on TRIM21-dependent protection of anti-NP antibodies, the in-vitro studies did not address the role and mechanism

of TRIM21.

Major points:

1. Fig 5. Was the CD4/CD8 T cell depletion assessed by flow cytometry on day 14 post-infection? In addition, the efficacy of depletion is suboptimal hampering the solidity of the conclusions that T cells don't play a role. Depletion efficacy should be presented in percentages and unless more than 95% of the cells are depleted, the conclusions cannot exclude the role of T cells in protection.
2. Fig 6 e and g. This experiment does not answer the question of NP antibody binding to Trim21 and preventing CCHFV infection. It is not surprising that the NP antibody reduced CCHFV replication when electroporated into the cell. This phenomenon is expected for any RNA virus NP antibody, as the NP antibody when electroporated into the cell will bind the NPL complex and prevent viral assembly.
3. As the authors mention in the discussion, NP is not on the surface of the virion or the infected cell. It is therefore unclear what the NP antibody binds to and how it enters the cytosol. To demonstrate the Fc-mediated function the authors should assess/answer the following questions:
 - Is TRIM21 or CCHFV NP expressed on the surface after infection
 - Does the NP antibody bind to TRIM21 in the absence of infection?
 - In vitro assessment using purified NP IgG or F(ab')₂ fragment
 - In vivo: Transfer of purified NP IgG or F(ab')₂ fragment
 - Demonstrate the effect of antibodies on infected cells without electroporation
 - The authors need to elucidate how the NP antibody can gain access to the nucleocapsid that is inside the viral envelope or the infected cells (WT & TRIM21 knockdown) . This would provide a mechanistic understanding to the manuscript. Without it, Fig 6e and 6g, electroporation data does not contribute to the suggested role of TRIM 21 in anti-NP dependent protection.

Minor points:

1. Introduction - CCHF was identified both in Crimea and in Congo, giving the virus its name. This should be corrected.
2. Introduction – CCHF transmission between ticks, livestock, and humans is not properly described and this should be corrected.
3. Introduction – Mentioning other antigens that confer protection is needed as NP is not the only protective antigen. It is important to mention other antigens to whom non-neutralizing antibodies were protective.
4. Why was the ELISA performed with lysed whole virion and not with the recombinant Nucleoprotein as performed in an earlier publication (PMID: 38382314)?
5. Discussion – Several works previously explored the role of TRIM21 in enveloped viruses including HBV, HCV, NDV, VSV,
6. The authors should better characterize the serum used for passive protection and determine the amount of IgG in both NP and sham serum samples.

Version 1:

Reviewer comments:

Reviewer #1

(Remarks to the Author)

The revised manuscript is significantly improved. The authors have sufficiently addressed my previous questions, and I congratulate the authors on a interesting story.

Reviewer #2

(Remarks to the Author)

Reviewer #3

(Remarks to the Author)

Thank you for this revision of the manuscript. It is good to see the authors have been able to extend their experiments to strengthen their conclusions. Overall the additions are valuable and changes made to the original manuscript are in line with review recommendations. The efforts to extend the T cell investigations are appreciated, and this reviewer looks forward to future work examining the effect of passively transferred antibodies or MAbs on T cell responses as a distinct response to that induced by vaccination.

There are only a few minor outstanding issues as follows:

Line 97 tracked changes document (line 93 main document) – is this an incorrect title?

Line 209 – please also reference 'Intracellular neutralization of rotavirus by VP6-specific IgG. PLoS Pathog 16(8):

Reviewer #4

(Remarks to the Author)

Leventhal et al. made significant efforts to answer the reviewers' concerns using multiple experimental approaches. The authors succeeded in demonstrating that CD8 T cells are not required for TRIM21-mediated NP antibody restriction of CCHF. Yet, despite these efforts, the mechanism underlying the protective role of TRIM21 and anti-NP has not been resolved.

Major points:

- Supplemental Figure 1: Surface staining of mAb 9D5 and NP immune sera in A549 infected cells look very different. The surface localization of NP is not consistent. Can the authors comment on this?
- Confocal staining or live cell imaging of infected A549 cells, staining for TRIM21, using the fluorescently tagged purified NP IgG from the sera could be done to look for co-localization of NP IgG and TRIM21 in infected/uninfected cells
- In Figure 7, high concentrations of NP antibodies in mouse and NHP and human sera have demonstrated a TRIM21-independent in vitro inhibition of CCHFV. In supplemental figure 7a, partial protection is seen in TRIM21 repNP vaccinated mice, suggesting that TRIM21 independent protection can occur. Hence the title of the paper is not accurate and should be changed from "requires TRIM21" to "TRIM21 plays a role in CCHFV protection...."
- In light of Supplemental Figure 1, It is unclear why the authors used the L929 cells and not the A549 cells that express NP on the surface for the invitro experiment in Figure 7. The authors should repeat this experiment with A549 cells.

We thank the reviewers for the comments and suggestions. We have addressed many comments and believe this has significantly improved the findings in the manuscript. Below please find a point-by-point response. Line numbers refer to the clean revised manuscript.

Reviewer #1 (Remarks to the Author):

In this manuscript, Leventhal et al. demonstrated that the protective effect of an alphavirus-based vaccine expressing CCHFV NP or passive transfer of NP-immune sera functions via TRIM21 receptor. This is an interesting study, which is of significance for improving our understanding towards how these “non-neutralizing” antibodies work. However, I also have some questions with the content.

1. A major point is the absence of mechanism studies to show how the sera enter cells (independent of Fcγ receptor) and subsequently bind to intracellular TRIM21 receptor. In addition, could the authors provide evidence of either NP antibodies or antibody-virus complex could bind with TRIM21?

We are actively investigating how antibodies enter cells but to-date we have not identified the mechanism. We have discussed this important question in the discussion (Lines 276 to 294). This will be the subject of future manuscripts. We now provide new data which shows that NP-immune sera can coordinate complex formation between NP and TRIM21 in the cytosol (Supplemental figure 6 and line 190 - 192).

2. They also didn't test the downstream events--how TRIM21 confer protection against CCHFV infection. As the authors discussed in the Discussion part, there are many possible mechanisms, but none of these has been tested. I strongly recommend the authors to detect the activation of innate immune response, inflammation related factors, apoptosis, or degradation of virions/viral proteins, which may provide possible explanations for how NP antibodies work. These can be done either in vivo or in TRIM21 knock out cell lines.

We agree and have in our resubmission performed innate immune activation assays. Using luciferase reporter assays and an immunocompetent CCHFV challenge model, we were unable to show activation of the innate immune response in cells electroporated with NP antibody and infected with CCHFV (supplemental figure 7d). Additionally, it has been shown that TRIM21 can expose viral genomes to RLRs activating the innate response. To test this in vivo, we found that our repNP conferred protection in MAVS^{-/-} mice indicating that NP antibody does not require the RLR-MAVS pathway to restrict CCHFV (Supplemental figure 7). While we cannot exclude the contribution of TLR sensing in this study, we consistently measure TRIM21-dependent phenotypes in mice with blocked type I IFN suggesting that innate immune activation is not necessary.

To address the degradation, we attempted to show degradation of NP via Western blot but were unsuccessful. We speculate that an amount of NP sufficient to show signal on a western blot is greater than the amount of NP that TRIM21 and antibody can coordinate to degrade. Further, when we attempted to perform our EDNA in the presence of epoxomicin, a specific and selective proteasome inhibitor, CCHFV replication was enhanced independently of antibody. Thus we were unable to conclude if epoxomicin relieved ab-mediated restriction of CCHFV. This suggests that the interaction of CCHFV with

the proteasome is complex and will require careful further study to understand the role of the proteasome in just CCHFV replication and in TRIM21-mediated restriction.

Lastly, we have repeated our EDNA with dilutions of sera and in WT and TRIM21 KO cells. We now present data that at high concentrations of antibody, TRIM21-independent restriction is possible suggesting that some protection may come from direct steric hindrance of interactions between NP and viral or host factors. It is unclear if this interaction confers protection *in vivo*, especially due to total lethality of CCHFV infection in TRIM21 KO mice (Figure 3). But for mouse, NHP and human sera, TRIM21 potentiates this restriction (Figure 7, line 163-201).

3. In Fig. 6, the authors showed intracellular NP antibodies could inhibit CCHFV infection by delivering antibodies into cells via electroporation. I understand the authors want to demonstrate once the antibodies are present in the cytoplasm, they could inhibit virus infection. However, I have some concerns: 1) This experiment was conducted independent of TRIM21 (TRIM21 knock out cell lines should be used), so I don't think it can well support their above findings. 2) they showed that in the absence of electroporation, NP antibodies could not enter cytoplasm, even in the presence of CCHFV. Therefore, again, I can't imagine that in the physiological conditions, how NP antibodies permeabilize into cells? I don't think this experiment could justify the *in vivo* protective effect of NP antibodies.

We agree with point 1 and have performed the studies in TRIM21 KO cells and at a range of dilutions with mouse, NHP and human sera. At increasing dilutions of sera, we see a potentiation of inhibition with a 17-fold, 3 and 6-fold increase in IC50 of sera across mouse, NHP and human sera respectively. Although this potentiation is modest, we believe that decreased IC50 and resistant fraction in absence of TRIM21 are likely the cause of death in lethally challenged mice. At high concentrations of antibody, we measured TRIM21-independent restriction of CCHFV with mouse and NHP sera, which we believe due to steric hindrance of NP interactions but this will require further study. However, even at the highest concentrations of sera used, in the absence of TRIM21, we measured a resistant fraction of virus suggesting that CCHFV can continue to replicate even in the presence of saturating amount of anti-NP antibody. This has been presented in a revised figure 7, line 163-201.

Point 2 is well taken and a key unanswered question of our study. However, our data cumulatively demonstrates that NP antibody and TRIM21 coordinate protection *in vivo* and our *in vitro* data show that this could be through direct restriction of CCHFV. It is possible this is a cell-type specific phenomenon as we show that NP is on the surface of infected A549 epithelial cells but not L929 fibroblasts. If NP is on the surface of infected cells, it is possible that NP-ab complexes are then internalized. But it is further unknown whether NP is on the surface of infected cells *in vivo*. Further, TRIM21 can be found on the cell surface and could recognize antibody-bound NP in the circulation. This is an area of active investigation and will be subject of future manuscripts. We have discussed these unanswered questions, line 276-294. Nevertheless, we believe our findings to be of significant value even though we have not identified if and how antibody reaches the cytoplasm of CCHFV target cells.

4. I don't agree with the statement "our data...suggested the term "non-neutralizing antibody" is inaccurate to describe antibodies against CCHFV NP" (Discussion part, the fourth line). NP antibodies are protective but not neutralizing because they can't neutralize virus infection in the classical neutralizing assay *in vitro*. Only when they were artificially electroporated into cells, virus infection was inhibited via unknown mechanism. Please also refer a recent publication "Nucleocapsid protein-specific monoclonal

antibodies protect mice against Crimean-Congo hemorrhagic fever virus (Garrison et al., Nature Communications, 2024)", which should be cited.

We agree that electroporation is an artificial way of delivering antibody into cells and have removed this statement. We have added discussion of this manuscript in the revised manuscript (Lines 247-252).

Reviewer #2 (Remarks to the Author):

Reviewer #3 (Remarks to the Author):

This manuscript presents a comprehensive study of the immunological mechanisms responsible for the efficacy of an alphavirus-based RNA vaccine expressing the CCHFV nucleoprotein. CCHFV is a high priority pathogen as listed by the WHO, and development of effective vaccination is therefore critical. In this study the authors examine the importance of a number of immune cell types, receptors and proteins, and show an important role for the intracellular receptor TRIM21.

Overall this is a very thorough and robust study, with extensive use of a challenging mouse model to explore numerous hypotheses. Virus quantification by both qPCR and TCID50 across multiple organs is appreciated as being particularly rigorous. The results show a clear and convincing role for TRIM21 in vaccine-mediated protection from CCHFV. However, the precise mechanism by which TRIM21 and antibody-NP co-localize is still unclear and the authors consider this well in the discussion and explain future questions that still need to be addressed.

The difference between protection mediated by vaccination versus passively transferred antibodies is intriguing. The theory that rapid TRIM21-mediated degradation of passively transferred NP antibody accounts for failure for complete protection is an interesting one. This could be explored further by quantifying how much NP antibody persists post infection. Blood sampling daily after infection and/or analysis of blood collected at the day 5 terminal cull could be achieved to address this.

Due to strict SOPs for samples coming out of the BSL4, we do not have an approved method for inactivating blood via a method that would leave antibody intact for ELISA quantification and we unfortunately did not collect serum at this timepoint. Additionally, we have previously seen significant amounts of anti-CCHFV IgG within 6 days of infection in mice so it is likely that at day 5 we would detect both remaining anti-NP antibody and early de novo responses. We do agree this is an intriguing question and we intend to explore the kinetics of antibody when treated with monoclonal antibodies in which we can alter the isotype to enable us to distinguish between mAb and host antibody response.

Alternatively, given the recently published study showing a role for CD8 T cells in TRIM21-dependent protection in LCMV infection, it would be interesting to see what the effect of CD8 T cell depletion has on passively transferred antibodies alone. Though the authors explore the role for T cells in vaccinated mice,

this is not explored for mice that receive passively transferred antibodies. The timing of when passively transferred mice start to recover from weight loss is remarkably similar to that observed in LCMV, which in turn coincides with induction of a CD8 T cell response in naïve animals. Though it is agreed that CD8 T cells were shown not to be important for vaccine-induced recall responses, this could be important for acute naïve CD8 T cell activation and protection mediated by future monoclonal antibody therapy.

We agree that excluding a role for T-cells is important. Although we did not explore with passive transfer mice due to lack of remaining sera to perform such study, we have now evaluated our repNP vaccine in CD8 KO mice. Cumulatively, we now have data showing that TRIM21 deficiency does not impact priming of T-cells in naïve or vaccinated mice after challenge, remains effective in mice depleted of CD4 and CD8 T-cells and remains effective in mice genetically deficient in cytotoxic T-cells (Figure 6 and lines 145-161). We do agree that it will be important to show in future studies developing the monoclonal antibody therapies that T-cells are not required but believe our current data comprehensively excludes a role for T-cells in vaccine mediated protection.

Minor comments:

Please explain what MAR1-5A3 is used for when it is first mentioned in the manuscript. **Added. Line 88.**

Figure 1: Error bars for first section of panel a? Label of x-axis of second panel needs updating to reflect IgM is in the panel. **The error bars are present in the first section of panel a but the data is very tight, so they are not visible. Updated the x-axis of the second panel**

Figure 2: Please clarify what is meant by ‘Immunology and challenge’ on day 0 (also in Fig 5). **This refers to groups of mice being either evaluated for immunological response to vaccination and necropsied prior to challenge or being challenged with lethal CCHFV and this is specified in the relevant figure legends.**

Figure 5: statistical significance for panel a? For all IFN γ ELISPOTs please define abbreviation SFC on y axis of these graphs. What does aCD4 or CD8 do to CCHFV viral loads in the absence of a vaccine? Was the sham aCD4/CD8 experiment not performed as mice die more rapidly? **The differences in panel a are not statistically significant, SFC abbreviation added to all relevant figure legends.**

We have previously shown that depletion of T-cells in naïve mice makes disease worse (Hawman et al 2021 and Rao et al 2023). However, in those studies, depletion of T-cells was evaluated in models where mice were expected to survive challenge. We did not include sham-vaccinated, T-cell depleted mice here to reduce numbers of mice used as we did not expect depletion of T-cells to improve disease.

Figure 6: Please provide clearer labelling for flow cytometry panel d, which sample is electroporated? Can NP-immune sera neutralize extracellularly? It is assumed not, but comparison of intracellular and extracellular neutralization capacity would be useful. This reviewer agrees that classical neutralizing assays are clearly limited for intracellular antibody activity, but the electroporation step in the intracellular neutralization assay makes it challenging to correlate this with true value in vivo. **Added extra labeling in now Supplemental Figure 5b and explanation in the figure legend to clarify flow cytometry. In previous publications covering development of the vaccine, NP sera was evaluated in**

classical neutralization assays in which the NP antibody would remain extracellular (Leventhal et al. 2022).

Discussion: TRIM21 has already been shown to be important for protection against the enveloped virus LCMV, so please update 'Our work extends the antiviral role of TRIM21 to enveloped viruses such as CCHFV'. A similar comment needs amending at the end of the discussion, but it is agreed that it is exciting to see another distinct enveloped virus protected by TRIM21.

We have added mention of this study in our discussion (Line 232-233), results (Lines 146-149) and modified the statements in the conclusion to clarify that we are not claiming this is first description of control of enveloped viruses (Line 304-305).

Reviewer #4 (Remarks to the Author):

Leventhal et al. study describes a TRIM21-mediated protective mechanism of Crimean-Congo Hemorrhagic Fever Virus (CCHFV) nucleoprotein (NP) antibodies induced by a self-replicating, alphavirus-based RNA vaccine that expresses NP. The authors first demonstrate partial protection of NP immune sera in passive transfer studies. Since the NP-specific antibodies were mostly of the IgG2c isotype, the authors assessed the Fc-mediated non-neutralizing function of the induced antibodies. All the WT, FcγR^{-/-}, C3^{-/-}, and NK cell-depleted mice vaccinated with the NP vaccine survived the lethal CCHFV challenge, with no significant difference in the viral loads. While comparable antibody responses in the TRIM21^{-/-} and WT vaccinated mice were observed, there was a complete loss of protection in the TRIM21^{-/-} mice. They claim NP vaccine-mediated protection is independent of T cells by performing a CD4/CD8 depletion study. The NP immune sera were not protective in a passive transfer study in TRIM21^{-/-} mice but were 75% protective in WT mice. Lastly, an in-vitro experiment suggested that the NP immune sera can significantly inhibit CCHFV replication when present inside the cytoplasm of the cell.

The study was overall well-designed. While the authors successfully demonstrate TRIM21-dependent protection of the NP antibodies induced by the vaccine, the Fc-mediated mechanism is not thoroughly demonstrated. In addition, as the paper focuses on TRIM21-dependent protection of anti-NP antibodies, the in-vitro studies did not address the role and mechanism of TRIM21.

Major points:

1. Fig 5. Was the CD4/CD8 T cell depletion assessed by flow cytometry on day 14 post-infection? In addition, the efficacy of depletion is suboptimal hampering the solidity of the conclusions that T cells don't play a role. Depletion efficacy should be presented in percentages and unless more than 95% of the cells are depleted, the conclusions cannot exclude the role of T cells in protection.

We measured T-cell depletion at day 5 post-infection. We achieved a 96.3% depletion of CD4 T-cells and 98.5% depletion of CD8 T-cells. We have also now evaluated our repNP vaccine in CD8 KO mice and showed complete protection. Cumulatively, we now have data showing that TRIM21 deficiency does not impact priming of T-cells in naïve or vaccinated mice after challenge, our vaccine remains effective in

mice depleted of CD4 and CD8 T-cells and remains effective in mice genetically deficient in cytotoxic T-cells (Figure 6). We believe this fully excludes a role of T-cells, particularly cytotoxic CD8 T-cells, in vaccine and antibody-mediated protection. Although we did not evaluate CD4 deficient mice as this would likely impair vaccine responses, we have previously shown that CD8 T-cells are the primary effector cell type in infected naïve mice (Rao et al. 2023) and thus residual CD4 T-cells are unlikely to play a role. However, as discussed for reviewer 3 these studies will be repeated as we develop monoclonal therapies against the CCHFV NP.

2. Fig 6 e and g. This experiment does not answer the question of NP antibody binding to Trim21 and preventing CCHFV infection. It is not surprising that the NP antibody reduced CCHFV replication when electroporated into the cell. This phenomenon is expected for any RNA virus NP antibody, as the NP antibody when electroporated into the cell will bind the NPL complex and prevent viral assembly.

We thank reviewer for this suggestion and have now performed the assay with dilution series of antibody and in WT and TRIM21 KO cells (new Figure 7 and Lines 163-201). At high concentrations of antibody we measured TRIM21-independent restriction of CCHFV we believe due to as reviewer suggests, steric hindrance of NP interactions. However, even at the highest concentrations of vaccinated mouse and NHP sera used, in the absence of TRIM21, we measured a resistant fraction of virus suggesting that CCHFV can continue to replicate even in the presence of saturating amount of anti-NP antibody. This was not seen with human sera suggesting there maybe qualitative differences in vaccine and infection induced anti-NP antibodies or the presence of antibodies against other viral proteins (mentioned Line 269-270). However, for both vaccine and infection induced NP antibody, at increasing dilutions of sera, we see a potentiation of inhibition with a 17-fold, 3 and 6-fold increase in IC50 of sera across mouse, NHP and human sera respectively (Figure 7). Although this potentiation is modest, we believe that decreased IC50 and resistant fraction in absence of TRIM21 are likely the cause of death in lethally challenged mice.

3. As the authors mention in the discussion, NP is not on the surface of the virion or the infected cell. It is therefore unclear what the NP antibody binds to and how it enters the cytosol. To demonstrate the Fc-mediated function the authors should assess/answer the following questions:

-Is TRIM21 or CCHFV NP expressed on the surface after infection: We have previously shown NP is not on the surface of mouse fibroblast L929 cells. But in light of the recent publication by Golden et al. 2024 exploring an NP mAb against CCHFV, we also evaluated human endothelial A549 cells. In these cells we did detect NP on the surface (New supplemental figure 1). Thus it is possible NP is on the cell surface but it may be cell type specific and is unclear how these findings translate in vivo. We did attempt to stain for TRIM21 with commercially available antibodies specific for human TRIM21 (we were unable to source mouse-specific reagents) and reported to be cross-reactive to mouse TRIM21 but got inconsistent results. One antibody stained only cytoplasmic while another stained both surface and cytoplasmic TRIM21. Due to these conflicting results, localization of TRIM21 and NP during CCHFV infection will require further investigation, likely generation of mouse-specific reagents and we believe beyond the scope of this current manuscript.

-Does the NP antibody bind to TRIM21 in the absence of infection? It is likely that NP antibody binds to TRIM21 in absence of infection but this is unlikely to activate TRIM21. Ubiquitination of the complex by TRIM21 appears to require formation of a dimer that in turn ubiquitinates a third TRIM21 molecule (Zeng et al 2021). This would likely only happen when multiple antibodies bound to their antigen are in close proximity. For example, in the study mentioned above, using a monoclonal antibody against monomeric GFP is unable to activate TRIM21. In contrast, polyclonal antibodies that can bind at multiple sites on monomeric GFP lead to engagement of multiple TRIM21 molecules on a single molecule and downstream activity. Alternatively, epitopes that occur in oligomers can cause TRIM21-mediated degradation by monoclonals since multiple abs in close proximity will bind to the oligomer. This later explanation is likely to be found in NP-coated incoming genomic segments. Thus while it is likely that NP antibody binds TRIM21 in absence of infection it is unlikely this would have a biological consequence until CCHFV infection and close interaction between multiple TRIM21 molecules occurs.

-In vitro assessment using purified NP IgG or F(ab')₂ fragment. We are actively developing monoclonal antibodies against NP and evaluating them in vivo for protection but currently do not have samples necessary to perform this analysis.

-In vivo: Transfer of purified NP IgG or F(ab')₂ fragment This will similarly be addressed by monoclonal antibodies in active development.

- Demonstrate the effect of antibodies on infected cells without electroporation In our classical neutralization assays, virus is mixed with serum and then added to cells. Thus our classical neutralization assays require the virus to grow in the presence of antibody for the 4 – 5 days it takes for CPE to develop. In these studies we did not see any effect of NP-immune sera (Leventhal et al. 2022). We agree that electroporation is artificially delivering antibody to the cytoplasm and are actively investigating how antibodies enter cells but to-date we have not identified the mechanism. We have discussed this important question in the discussion (Lines 276 to 294). This will be the subject of future manuscripts.

- The authors need to elucidate how the NP antibody can gain access to the nucleocapsid that is inside the viral envelope or the infected cells (WT & TRIM21 knockdown) . This would provide a mechanistic understanding to the manuscript. Without it, Fig 6e and 6g, electroporation data does not contribute to the suggested role of TRIM 21 in anti-NP dependent protection.

We agree that electroporation is an artificial way of delivering antibody into cells and determining how this occurs in vivo is a key unanswered question (Lines 276 to 294). This is an area of active investigation. We have added data showing that NP may be on the surface of some cells but this will require *in vivo* investigation. However, even in the absence of this mechanism our manuscript provides important, novel insight into how antibodies can control viral infections, the role of TRIM21 in antiviral immunity and mechanistic insight into how a multitude of vaccines for CCHFV moving towards clinical trials are likely to protect. To our knowledge, at least two NP-based vaccines are soon to enter human clinical trials and our findings identify the mechanisms that should be evaluated to predict how effective these vaccines would be in preventing CCHF. In addition, we show with human sera that antibody arising during natural infection can restrict CCHFV similarly to vaccine elicited antibody (Figure 7), suggesting our findings may

extend beyond vaccine-elicited immunity to natural immunity as well. In addition, an independent group showed significant protection when mice were treated with an anti-NP monoclonal (Golden et al 2024) further supporting our findings. Therefore we believe that although we have not identified how or where NP antibody accesses the CCHFV NP *in vivo* our findings are of sufficient importance for publication. The mechanism by which these antibodies access NP is an area of active investigation and will be the subject of future manuscripts.

Minor points:

1. Introduction - CCHF was identified both in Crimea and in Congo, giving the virus its name. This should be corrected. **Corrected. Line 35-36**
2. Introduction – CCHF transmission between ticks, livestock, and humans is not properly described and this should be corrected. **We have clarified how humans are infected with CCHFV, Line 41-43 but transmission of CCHFV and how it is maintained in nature via tick and amplifying hosts is beyond the scope of our introduction.**
3. Introduction – Mentioning other antigens that confer protection is needed as NP is not the only protective antigen. It is important to mention other antigens to whom non-neutralizing antibodies were protective. **We have added a statement that abs to other antigens are protective. Line 64-65.**
4. Why was the ELISA performed with lysed whole virion and not with the recombinant Nucleoprotein as performed in an earlier publication (PMID: 38382314)? **In our experience the lysed whole-virion ELISA is more sensitive than the recombinant antigen ELISA while also being cheaper. And given that the mice received only the repNP we did not feel it necessary to use a specific recombinant antigen as there were no other viral antigens present to distinguish from.**
5. Discussion – Several works previously explored the role of TRIM21 in enveloped viruses including HBV, HCV, NDV, VSV, **To our knowledge the role of TRIM21 in these viral infections does not require the presence of antibody. In the interest of space, we have clarified in the discussion that our findings extend the role of antibody and TRIM21 in control of enveloped viruses (Line231-233).**
6. The authors should better characterize the serum used for passive protection and determine the amount of IgG in both NP and sham serum samples. **repNP sera was quantified to have ~5.03mg/mL IgG, this information is added to the first results section describing the serum Line 85.**

Reviewer #1 (Remarks to the Author):

The revised manuscript is significantly improved. The authors have sufficiently addressed my previous questions, and I congratulate the authors on a interesting story.

We thank the reviewer for their comments and suggestions.

Reviewer #2 (Remarks to the Author):

We thank the reviewer for their comments and suggestions.

Reviewer #3 (Remarks to the Author):

Thank you for this revision of the manuscript. It is good to see the authors have been able to extend their experiments to strengthen their conclusions. Overall the additions are valuable and changes made to the original manuscript are in line with review recommendations. The efforts to extend the T cell investigations are appreciated, and this reviewer looks forward to future work examining the effect of passively transferred antibodies or MAbs on T cell responses as a distinct response to that induced by vaccination.

We thank the reviewer for their comments and suggestions.

There are only a few minor outstanding issues as follows:

Line 97 tracked changes document (line 93 main document) – is this an incorrect title? Yes, we have fixed this. (Line 93)

Line 209 – please also reference ‘Intracellular neutralization of rotavirus by VP6-specific IgG. PLoS Pathog 16(8): e1008732. doi: 10.1371/journal.ppat.1008732’ where this assay was first described. Added this reference (line 167).

Reviewer #4 (Remarks to the Author):

Leventhal et al. made significant efforts to answer the reviewers’ concerns using multiple experimental approaches. The authors succeeded in demonstrating that CD8 T cells are not required for TRIM21-mediated NP antibody restriction of CCHF. Yet, despite these efforts, the mechanism underlying the protective role of TRIM21 and anti-NP has not been resolved.

Major points:

- Supplemental Figure 1: Surface staining of mAb 9D5 and NP immune sera in A549 infected cells look very different. The surface localization of NP is not consistent. Can the authors comment on this? One potential explanation is due to the epitopes targeted by 9D5 versus polyclonal sera. MAb 9D5 recognizes a specific epitope within the stalk region of NP (Garrison et al. 2024). We have not performed epitope

mapping of repNP-elicited sera but it may target different epitopes that are differentially distributed.

- Confocal staining or live cell imaging of infected A549 cells, staining for TRIM21, using the fluorescently tagged purified NP IgG from the sera could be done to look for co-localization of NP IgG and TRIM21 in infected/uninfected cells. We investigated co-localization of TRIM21, NP, and NP antibody using the SmBiT/LgBiT system (supplemental figure 6), where luminescence is only achieved by these proteins closely interacting. This assay was done using NP-immune serum and pre-immune serum to show that this interaction is NP antibody specific. Confocal imaging likely would require large amounts of protein to achieve sufficient signal for detection and based on experience would likely saturate the endogenous amount of TRIM21 resulting in most NP/IgG failing to colocalize. This contrasts with how we speculate the mechanism works during infection where on initial infection relatively few viral particles bind to a cell, delivering relatively small amounts of NP that are then recognized by IgG and TRIM21. This in turn blocks infection.

- In Figure 7, high concentrations of NP antibodies in mouse and NHP and human sera have demonstrated a TRIM21-independent *in vitro* inhibition of CCHFV. In supplemental figure 7a, partial protection is seen in TRIM21 repNP vaccinated mice, suggesting that TRIM21 independent protection can occur. Hence the title of the paper is not accurate and should be changed from “requires TRIM21” to “TRIM21 plays a role in CCHFV protection...”

We agree that our data demonstrate TRIM21 independent protection, particularly with the MA-CCHFV model. This likely reflects the less stringent challenge with this model as the virus is homologous to the vaccine antigen and the MAR1-5A3 UG3010 model requires much greater control of the challenge virus for survival. However, we still feel that TRIM21 plays an integral role in the protection observed *in vivo* and have changed the title to “Antibodies Targeting the CCHFV Nucleoprotein via TRIM21”

- In light of Supplemental Figure 1, It is unclear why the authors used the L929 cells and not the A549 cells that express NP on the surface for the *in vitro* experiment in Figure 7. The authors should repeat this experiment with A549 cells. We developed this assay based on established protocols using L929 cells (line 167) a cell type we knew CCHFV to grow well in and in which we were able to access well characterized TRIM21 knock-out cells. Since this manuscript is largely focused on our mouse model of CCHFV we feel mouse L929 cells were appropriate to answer our questions. We do not believe repeating the studies in human A549 cells would alter our conclusions as TRIM21 is evolutionarily conserved from mice to humans (Keeble et al. 2008). This is also evidenced by TRIM21-dependent restriction by human sera in mouse cells demonstrating that mouse TRIM21 can engage human IgG. Precisely where TRIM21, NP and anti-NP antibodies colocalize is the subject of active investigation and will likely require *in vivo* confirmation of any *in vitro* findings.

The sera samples, particularly the human and NHP samples used in these studies are also highly limited and are being utilized for additional studies. However, we have evaluated the cynomolgus sera presented here in a bead-based assay in which human TRIM21 is used as a detector. With this assay we have shown that NP-specific antibody from vaccinated cynomolgus macaques can bind human TRIM21 (see below). This supports a hypothesis that our findings are not limited to mice. However, this data is being

incorporated to a separate manuscript in preparation and will not be included in this manuscript.

[Redacted]